# Numerical coupling of aerosol emissions, dry removal, and turbulent mixing in the E3SM Atmosphere Model version 1 (EAMv1), part II: a semi-discrete error analysis framework for assessing coupling schemes

Christopher J. Vogl[1], Hui Wan[2], Carol S. Woodward[1], and Quan M. Bui[1,*]

[1]Center for Applied Scientific Computing, Lawrence Livermore National Laboratory, Livermore, California, USA
[2]Atmospheric, Climate, and Earth Sciences Division, Pacific Northwest National Laboratory, Richland, Washington, USA
[*]Now at: Blue River Technology, Sunnyvale, California, USA

**Correspondence:** Christopher J. Vogl (vogl2@llnl.gov)

**Abstract.** Part I of this study discusses the motivation and empirical evaluation of a revision to the aerosol-related numerical process coupling in the atmosphere component of the Energy Exascale Earth System Model version 1 (EAMv1) to address the previously reported issue of strong sensitivity of the simulated dust aerosol lifetime and dry removal rate to the model's vertical resolution. This paper complements that empirical justification of the revised scheme with a mathematical justification

leveraging a semi-discrete analysis framework for assessing the splitting error of process coupling methods. The framework distinguishes the error due to numerical splitting from the error due to the time integration method(s) used for each individual process. Such a distinction results in a framework that provides an intuitive understanding of the causes of the splitting error. The application of this framework to dust life cycle in EAMv1 confirms (i) that the original EAMv1 scheme artificially strengthens the effect of dry removal processes, and (ii) that the revised splitting reduces that artificial strengthening.

While the error analysis framework is presented in the context of the dust life cycle in EAMv1, the framework can be broadly leveraged to evaluate process coupling schemes, both in other physical problems and for any number of processes. This framework will be particularly powerful when the various process implementations support a variety of time integration approaches. Whereas traditional local truncation error approaches require separate consideration of each combination of time integration methods, this framework enables evaluation of coupling schemes independent of particular time integration approaches for

each process while still allowing for the incorporation of these specific time integration errors if so desired. The framework also explains how the splitting error terms result from (i) the integration of individual processes in isolation from other processes and (ii) the choices of input state and timestep size for the isolated integration of processes. Such a perspective has the potential for rapid development of alternative coupling approaches that utilize knowledge both about the desired accuracy and about the computational costs of individual processes.

## 1  Introduction

Accurate representation of process interactions is an important and ubiquitous challenge in multiphysics modeling (Keyes et al., 2013). For the sake of tractability, splitting methods are widely used that allow for the separate development of both the continuum and discrete representation of individual processes, which are then assembled to form a multi-process numerical model. In weather, climate, and Earth system modeling, it has been recognized that how to combine the different process representations to form a coherent and accurate numerical model is a challenge deserving more attention, see the review paper by Gross et al. (2018) and the references therein, as well as more recent studies by, e.g., Barrett et al. (2019), Donahue and Caldwell (2020), Wan et al. (2021), Santos et al. (2021), Ubbiali et al. (2021), and Zhou and Harris (2022). The dust aerosol life cycle problem discussed in the companion paper (Part I,  Wan et al., 2023) is a recent example from the atmosphere component of the Energy Exascale Earth System Model version 1 (EAMv1, Rasch et al., 2019) that shows that different numerical methods used for process coupling in the overall time integration can lead to substantially different results at a fixed spatial resolution, as well as significantly different sensitivities to spatial resolution change.

That companion paper clarified the main source and sink processes in the dust life cycle in EAMv1, quantified their relative magnitudes, reflected on the process coupling scheme used in the default model, proposed a revised coupling scheme, and evaluated the impact of the revised coupling on the simulated aerosol climatology. The discussions therein are based primarily on the intuition of atmospheric modelers, and the reasoning was verified by confirming agreement between the expected and obtained numerical results from EAMv1. To gain more confidence that the EAMv1 solution obtained with the revised coupling scheme is indeed a better numerical solution, i.e., closer to the true or trusted solution than that of the original EAMv1 scheme, a mathematical explanation for the changes with the revised coupling is needed. Typically, computational analyses are done to develop such explanations, including timestep self-convergence studies, such as those in Wan et al. (2013), or timestep sensitivity studies, such as those in Wan et al. (2021) and Santos et al. (2021). Both of these approaches are unfortunately impractical in this case, because in the current EAM code, the coupling timestep for dust emissions, dry removal, and turbulent mixing is tied to the coupling timesteps of various other atmospheric processes, such as aerosol microphysics and gas-phase chemistry, deep convection and aerosol wet removal, and the coupling between the resolved dynamics and the parameterizations. Thus, without significant code structure changes, it is not feasible to isolate the impact of coupling approaches for the three aerosol processes we would like to focus on.

This paper provides a theoretical explanation of the numerical results presented in Part I and introduces a framework based on truncation error analysis that can more broadly help address the research gap in numerical process modeling in weather, climate, and Earth system modeling. In a fully discretized model with process splitting, the overall local truncation error from time integration includes contributions from (i) the time integration of each individual process and (ii) the splitting of each process from the remaining processes. In the atmosphere modeling literature, there are a number of theoretical studies that leverage truncation error analysis to compare the accuracy of different splitting methods (e.g., Caya et al., 1998; Staniforth et al., 2002; Dubal et al., 2004, 2005, 2006; Ubbiali et al., 2021). Considering the added complexity of the weather, climate, and Earth system models, especially the long-term and large-team efforts that are typically needed for the continuous development

of such models, it is useful to understand and reduce the different types of error separately. Because the numerical results in Part I showed substantial sensitivity to the process coupling approach, this work develops a framework that leverages exact time integration of the processes to focus on understanding and reducing the splitting error. Because the time discretization is also an error source worth assessing and addressing, the framework is designed to be flexible enough to incorporate the individual process time integration errors (see discussion at the end of Sect. 2.4).

In the work of Williamson (2013) that discusses issues related to atmospheric convection and process splitting, exact time integration is also used for individual processes in a two-process equation and a three-process equation. However, the purpose there was to point out a problem in the formulation of a specific parameterization; hence and understandably, that work did not use exact time integration as a general tool for analyzing splitting errors in physics systems beyond the two highly idealized and customized equations discussed therein. In this paper, we demonstrate the usage of exact time integration within an error analysis framework that distinguishes splitting error and the error resulting from temporal discretization of individual processes, providing an approach that allows for focus on the splitting error in general problems involving two or more processes. While such a focus is commonplace in the field of mathematics (e.g., Hairer et al., 2006; LeVeque, 1982), the approach has not caught much attention in the weather and climate modeling communities despite the important benefits that its adoption can provide to the development of sophisticated atmospheric numerical models.

For example, in EAM and its predecessors, changes in time integration methods have been implemented both in the dynamical core representing the resolved fluid dynamics and in various parameterizations representing sub-grid-scale processes. The results from a local truncation error analysis that did not distinguish between splitting and time integration sources would become invalid as soon as any time integration method was changed, hence separate considerations would be needed for every combination of time integration methods used by the dynamical core and the many physics parameterizations. In contrast, the results from an error analysis approach that can consider splitting error independent of time integration sources are expected to have a better chance of remaining valid across multiple versions of the same atmosphere model and might even generalize to other atmosphere models.

Another benefit of the splitting error analysis framework demonstrated in this work is that the terms in error expressions produced are easily attributed to the coupling choices made for the individual right-hand-side (RHS) terms of the continuum equations. In atmosphere modeling terminology, the framework goes beyond deriving the splitting errors that contribute to the overall error in prognostic variables: it also demonstrates how the coupling choices lead to errors in the process rates (i.e., rates of change of prognostic variables, also referred to as tendencies) associated with the various physical processes. For this work, coupling choices for the dust source and sink processes are directly mapped to the splitting error terms that contribute to overall error in the EAM-simulated mixing ratios. Reducing splitting errors in the process rates can help avoid compensating errors from different physical processes and, thus, help ensure the model provides good predictions of the prognostic variables for the right reasons. Additionally, understanding the impact of numerical coupling scheme choices at the process level allows for the development of new coupling strategies that focus on improving the accuracy of numerical process rates, and the associate prognostic variables, while considering the computational cost of the various processes.

Whereas the companion paper (Wan et al., 2023) focuses on motivating the dust life cycle problem and empirical comparison of two coupling methods in consideration, this paper focuses on how a semi-discrete error analysis supports the empirical finding that one coupling method leads to a better numerical solution than the other. This two-part approach facilitates a detailed description of the framework with significant pedagogical values to weather and climate model developers. As an example, our own collaboration between applied mathematicians and atmospheric scientists on the investigation of dust life cycle in EAMv1 has shown that a step-by-step explanation of the derivation of splitting errors resulting from two coupling methods, which are widely used in weather and climate models, was helpful for the EAM developers in this collaboration to recognize the relevance, as well as the generality, of the semi-discrete methodology. In addition, one of the points we make in Sect. 3 is that after splitting errors are derived for coupling schemes used in two-process problems, it is possible to use those error expressions as building blocks to perform back-of-the-envelope derivations for problems involving more processes, making the derivations much less tedious and the framework much easier to use by researchers of specific applications. The discussion in Sect. 3.2 can be viewed as an example of such a back-of-the-envelope derivation, demonstrating that the mathematically rigorous framework can be made accessible to physical scientists working on practical problems. Given the reinvigorated interests in numerical process coupling reflected in the review by Gross et al. (2018) and the community efforts described in, e.g., Heinzeller et al. (2023), the pedagogical description of the error analysis framework presented here can be a useful contribution to those model development efforts.

To focus on the error analysis framework, the remainder of this paper forgoes the background on the motivating dust life cycle problem and coupling methods in consideration, for which the reader is referred to the Part I work (Wan et al., 2023), and instead opens by introducing the analysis framework in Sect. 2 using a generic two-process problem and deriving the splitting errors associated with the widely used parallel and sequential splitting methods. Section 3 then applies the error analysis framework to a three-process problem inspired by the dust aerosol life cycle in EAMv1. Leading-order splitting errors are derived both for the original process coupling in EAMv1 and for the revision proposed in Part I, and the characteristics of the splitting errors are discussed. This paper concludes by summarizing results in Sect. 4. An appendix gives mathematical details of the analysis framework.

## 2   A semi-discrete analysis framework for assessing splitting error of process coupling methods

The error analysis framework demonstrated in this work is described as semi-discrete in that it takes the perspective that while the overall time integration of the model is discrete, the integration of individual processes is done exactly (i.e., there is no temporal discretization for each process). This perspective is the critical component that allows the framework to isolate the splitting truncation error from that of the process temporal discretization errors. The semi-discrete approach allows for the derivation of splitting truncation errors from how the coupling scheme incorporates individual process. By casting the numerical splitting of processes as estimating process rates in split fashion, the framework identifies two general coupling method choices that cause splitting truncation errors. In Sect. 2.2, those coupling choices are identified in two widely-used coupling methods for two-process problems. In Sect. 2.3, the splitting truncation error terms that result from those choices are

derived and combined to form the leading-order splitting truncation error for each coupling method. In Sect. 2.4, the terms in the leading-order splitting truncation error are attributed back to the coupling method choices in a manner that (i) identifies when the splitting truncation errors from coupling methods compound or cancel each other and (ii) provides a workflow to easily generalize the results to problems with more than two processes.

## 2.1 Notation and definitions

Consider a prognostic equation, with multiple operators, that can either be an ordinary differential equation (ODE) or a partial differential equation (PDE). Noting that the method-of-lines will reduce a PDE to an ODE, the prognostic equation to be studied herein is introduced in ODE form:

$$\frac{dq(t)}{dt} = \sum_{i=1}^{I} X_i\big(q(t)\big), \; I > 1, \; t > 0, \; q(0) = q^{\mathrm{IC}}, \tag{1}$$

where the different processes, $X_i$, are discretized and implemented by different components of the model software (i.e., the processes are split). At time $t_n$, denote $q(t_n)$ and $q^n$ as the exact and numerical (approximate) solutions, respectively. The error at time $t_n$ is defined as

$$\mathcal{E}^n \equiv q^n - q(t_n).$$

Let $\mathcal{F}_{\Delta t}(q^{\mathrm{input}})$ represent the numerical algorithm that advances the solution from state $q^{\mathrm{input}}$ to state $q^{\mathrm{output}}$, where $\Delta t$ is the timestep size used. In other words, denote $\mathcal{F}_{\Delta t}$ as the mapping such that the numerical solution at $t_{n+1} = t_n + \Delta t$ is

$$q^{n+1} = \mathcal{F}_{\Delta t}(q^n).$$

The solution error at time $t_{n+1}$ can be expressed as

$$\begin{aligned} \mathcal{E}^{n+1} &= \mathcal{F}_{\Delta t}(q^n) - q(t_{n+1}) \\ &= \mathcal{F}_{\Delta t}(q^n) - \mathcal{F}_{\Delta t}\big(q(t_n)\big) + \mathcal{F}_{\Delta t}\big(q(t_n)\big) - q(t_{n+1}) \\ &= \underbrace{\frac{d\mathcal{F}_{\Delta t}}{dq}(\gamma_n)\mathcal{E}^n}_{\text{propagated error}} + \underbrace{\mathcal{F}_{\Delta t}\big(q(t_n)\big) - q(t_{n+1})}_{\text{local truncation error } (lte)}, \end{aligned} \tag{2}$$

where $\gamma_n$ is a value of $q$ between $q(t_n)$ and $q^n$ by the mean value theorem. The error $\mathcal{E}^{n+1}$ consists of the evolution of the existing error at time $t_n$ (the propagated error) and the generation of new error from time $t_n$ to $t_{n+1}$ (the local truncation error).

If each aforementioned component of the model software implements a single process in Eq. (1) without using information about other processes, then the role of such a component of the software can be interpreted as integrating the following one-process ODE

$$\frac{dq_{X_i}(t)}{dt} = X_i\big(q_{X_i}(t)\big), \; t > t_n, \; q_{X_i}(t_n) = q_{X_i}^{\mathrm{input}}. \tag{3}$$

Depending on which coupling scheme is used, $q_{X_i}^{\text{input}}$ can be the numerical solution of the multi-process problem at $t_n$ (i.e., $q^n$) or some value of the physical quantity, $q$, passed to the component of the software, $X_i$, by another component of the software, $X_j$. In the following, the notation $q_{X_i}[t - t_n; q_{X_i}^{\text{input}}]$ is used to denote the exact solution of the one-process ODE (3), namely,

$$
q_{X_i}[t - t_n; q_{X_i}^{\text{input}}] \equiv q_{X_i}(t)
$$

$$
= q_{X_i}^{\text{input}} + \int_{t_n}^{t} X_i\big(q_{X_i}(\eta)\big)\, d\eta. \tag{4}
$$

The explicit mentioning of $q_{X_i}^{\text{input}}$ in the square brackets on the left side of Eq. (4) emphasizes the dependence of $q_{X_i}$ and $X_i$ on $q_{X_i}^{\text{input}}$, the significance of which will become clear below. Using a similar notation for the exact solution of the multi-process problem Eq. (1), one can write

$$
q[t - t_n; q(t_n)] \equiv q(t)
$$

$$
= q(t_n) + \int_{t_n}^{t} \sum_{i=1}^{I} X_i\big(q(\eta)\big)\, d\eta. \tag{5}
$$

To facilitate comprehension of the derivations below, it is again emphasized that the time integrals in Eq. (4) and Eq. (5) are assumed to be exactly evaluated. It is also useful to note that, by definition,

$$
q_{X_i}[0; q_{X_i}^{\text{input}}] = q_{X_i}^{\text{input}}, \quad q[0; q(t_n)] = q(t_n). \tag{6}
$$

## 2.2 Sources of splitting error

From Eq. (5), one can see the average process rate for $X_i$ that contributes to the change in $q$ from $t_n$ to $t_n + \Delta t$ in the original multi-process ODE is

$$
\frac{1}{\Delta t} \int_{t_n}^{t_n + \Delta t} X_i\big(q(\eta)\big)\, d\eta
$$

$$
= \frac{1}{\Delta t} \int_{t_n}^{t_n + \Delta t} X_i\big(q[\eta - t_n; q(t_n)]\big)\, d\eta
$$

$$
= \frac{1}{\Delta t} \int_{0}^{\Delta t} X_i\big(q[\tilde{\eta}; q(t_n)]\big)\, d\tilde{\eta},
$$

while the $X_i$ process considered in isolation using Eq. (4) results in a approximation to the average process rate of

$$
\frac{1}{\Delta t} \int\limits_{t_n}^{t_n+\Delta t} X_i\Big(q_{\mathrm{x}_i}(\eta)\Big)\, d\eta
$$

$$
= \frac{1}{\Delta t} \int\limits_{t_n}^{t_n+\Delta t} X_i\Big(q_{\mathrm{x}_i}[\eta - t_n; q_{\mathrm{x}_i}^{\mathrm{input}}]\Big)\, d\eta
$$

$$
= \frac{1}{\Delta t} \int\limits_{0}^{\Delta t} X_i\Big(q_{\mathrm{x}_i}[\tilde{\eta}; q_{\mathrm{x}_i}^{\mathrm{input}}]\Big)\, d\tilde{\eta}.
$$

The discrepancy between these two average process rates has two sources:

1. The function $q_{\mathrm{x}_i}$ differs from the function $q$, as the time evolution of $q_{\mathrm{x}_i}$ is controlled by a single process (see Eq. (3)) while the evolution of $q$ is controlled by multiple processes (see Eq. (1)).

2. The input state $q_{\mathrm{x}_i}^{\mathrm{input}}$ used for integrating the equation of process $X_i$ can differ from $q(t_n)$.

In other words, the error in the estimated process rate $X_i$ (or increment $X_i \Delta t$) can arise from (1) treating the process in
isolation without considering the influence of other processes on the physical quantity, $q$, and (2) starting the single-process integration with an input that deviates from the solution of the multi-process ODE. These two types of error are referred to as *isolation-induced error* and *input-induced error*, respectively, in the remainder of the paper. To further elaborate on these two sources of splitting error, consider the following generic two-process ODE

$$
\frac{dq(t)}{dt} = A\big(q(t)\big) + B\big(q(t)\big),\ t > 0,\ q(0) = q^{\mathrm{IC}}, \tag{7}
$$

which has the following exact solution at $t_{n+1}$, written in terms of the exact solution at $t_n$:

$$
q(t_{n+1}) = q(t_n) + \int\limits_{0}^{\Delta t} A\big(q[\eta; q(t_n)]\big) + \int\limits_{0}^{\Delta t} B\big(q[\eta; q(t_n)]\big)\, d\eta
$$

Below, the errors of two widely used coupling schemes are analyzed: parallel and sequential splitting, both of which are depicted in Fig. 1 using a flowchart description, a pseudo-code description, and an ODE description.

**Sources of error in parallel splitting**

A parallel splitting scheme first lets each model component estimate the process rate of a single process and then sums up the corresponding increments to advance an input value of $q$ to an output value of $q$. The scheme can be represented by a mapping $q^{\mathrm{output}} = \mathcal{F}_{\Delta t}^{\mathrm{PS}}(q^{\mathrm{input}})$ where

$$
\mathcal{F}_{\Delta t}^{\mathrm{PS}}(q^{\mathrm{input}}) \equiv q^{\mathrm{input}} + \Delta t\,(A^* + B^*)\,,
$$

$$
A^* \equiv \Big(q_A[\Delta t; q^{\mathrm{input}}] - q^{\mathrm{input}}\Big)/\Delta t\,,
$$

$$
B^* \equiv \Big(q_B[\Delta t; q^{\mathrm{input}}] - q^{\mathrm{input}}\Big)/\Delta t\,,
$$

**(a1) Parallel splitting, flowchart description**

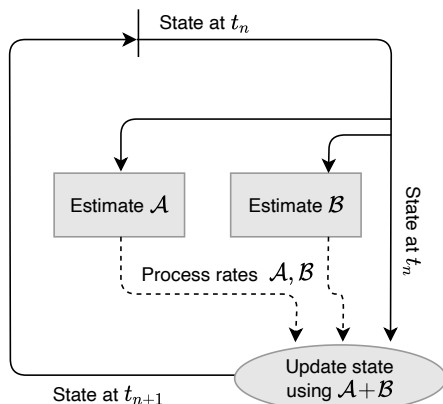

**(a2) Sequential splitting, flowchart description**

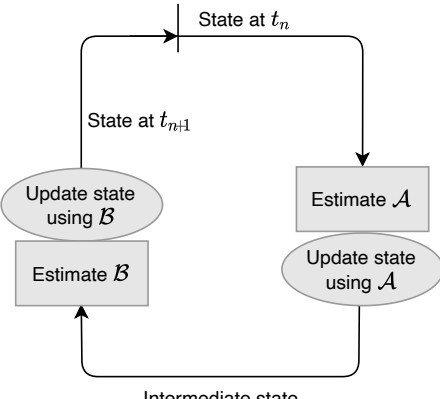

**(b1) Parallel splitting, pseudo code description**

$$q^n \xrightarrow{\text{apply } A \text{ process for duration } \Delta t} q_A^{n+1}; \quad \text{let } A^* = \frac{q_A^{n+1} - q^n}{\Delta t}.$$

$$q^n \xrightarrow{\text{apply } B \text{ process for duration } \Delta t} q_B^{n+1}; \quad \text{let } B^* = \frac{q_B^{n+1} - y^n}{\Delta t}.$$

$$\text{Let } q^{n+1} = q^n + \Delta t(A^* + B^*).$$

**(b2) Sequential splitting, pseudo code description**

$$q^n \xrightarrow{\text{apply } A \text{ process for duration } \Delta t} q_A^{n+1}.$$

$$q^A \xrightarrow{\text{apply } B \text{ process for duration } \Delta t} q_B^{n+1}.$$

$$\text{Let } q^{n+1} = q_B^{n+1}.$$

**(c1) Parallel splitting, ODE description**

1. Solve for $q_A(t_{n+1})$ given $\dfrac{dq_A}{dt} = A(q_A)$ with $q_A(t_n) = q^n$.

2. Solve for $q_B(t_{n+1})$ given $\dfrac{dq_B}{dt} = B(q_B)$ with $q_B(t_n) = q^n$.

3. Let $A^* = \dfrac{q_A^{n+1} - q^n}{\Delta t}$ and $B^* = \dfrac{q_B^{n+1} - q^n}{\Delta t}$.

4. Let $q^{n+1} = q^n + \Delta t(A^* + B^*)$.

**(c2) Sequential splitting, ODE description**

1. Solve for $q_A(t_{n+1})$ given $\dfrac{dq_A}{dt} = A(q_A)$ with $q_A(t_n) = q^n$.

2. Solve for $q_B(t_{n+1})$ given $\dfrac{dq_B}{dt} = B(q_B)$ with $q_B(t_n) = q_A^{n+1}$.

3. Let $q^{n+1} = q_B^{n+1}$.

**Figure 1.** The parallel splitting (left column) and sequential splitting method (right column) for solving the two-process ODE defined in Eq. (7). The top, middle, and bottom rows depict the methods in three different ways.

which gives

$$\mathcal{F}^{\text{PS}}_{\Delta t}(q^{\text{input}}) = q^{\text{input}} + \int_0^{\Delta t} A\Big(q_A[\eta; q^{\text{input}}]\Big) d\eta$$

$$+ \int_0^{\Delta t} B\Big(q_B[\eta; q^{\text{input}}]\Big) d\eta. \tag{8}$$

Recall that the local truncation error for a numerical method is, per the definition given in Eq. (2), the difference between the exact solution at $t_{n+1}$ and the solution at $t_{n+1}$ obtained from the numerical method applied to the exact solution at $t_n$. Thus, the local truncation error for parallel splitting is the difference between the exact two-process solution given by Eq. (5) and the result of Eq. (8) with $q^{\text{input}} = q(t_n)$:

$$\mathcal{F}^{\text{PS}}_{\Delta t}\Big(q(t_n)\Big) - q(t_{n+1})$$

$$= \underbrace{\int_0^{\Delta t} A\Big(q_A[\eta; q(t_n)]\Big) d\eta - \int_0^{\Delta t} A\Big(q[\eta; q(t_n)]\Big) d\eta}_{lte^{\text{PS}}_A}$$

$$+ \underbrace{\int_0^{\Delta t} B\Big(q_B[\eta; q(t_n)]\Big) d\eta - \int_0^{\Delta t} B\Big(q[\eta; q(t_n)]\Big) d\eta}_{lte^{\text{PS}}_B}. \tag{9}$$

Note that because the time integration for $A$ and $B$ both start from $q(t_n)$, there are only local truncation errors caused by treating $A$ and $B$ in isolation (i.e., no input-induced error). Also note that because parallel splitting treats $A$ and $B$ the same way, the local truncation error expression shows symmetry between the two processes.

**Sources of error in sequential splitting**

The sequential splitting scheme discussed here handles different processes in a successive manner, letting each model compartment operate on an input value of $q$ and return an updated value of $q$. Here, operator $A$ is evaluated first, and the result is

then used as input to operator $B$. The method can be represented by a mapping $q^{\text{output}} = \mathcal{F}^{\text{SS}}_{\Delta t}(q^{\text{input}})$ where

$$\mathcal{F}^{\text{SS}}_{\Delta t}(q^{\text{input}}) \equiv q_B\left[\Delta t; q_A[\Delta t; q^{\text{input}}]\right] \tag{10}$$

$$= q_A[\Delta t; q^{\text{input}}] + \int_0^{\Delta t} B\left(q_B\left[\eta; q_A[\Delta t; q^{\text{input}}]\right]\right) d\eta$$

$$= q^{\text{input}} + \int_0^{\Delta t} A\left(q_A[\eta; q^{\text{input}}]\right) d\eta$$

$$+ \int_0^{\Delta t} B\left(q_B\left[\eta; q_A[\Delta t; q^{\text{input}}]\right]\right) d\eta. \tag{11}$$

The local truncation error for sequential splitting is the difference between the exact two-process solution given by Eq. (5) and the result of Eq. (11) with $q^{\text{input}} = q(t_n)$:

$$\mathcal{F}^{\text{SS}}_{\Delta t}\left(q(t_n)\right) - q(t_{n+1})$$

$$= \underbrace{\int_0^{\Delta t} A\left(q_A[\eta; q(t_n)]\right) d\eta - \int_0^{\Delta t} A\left(q[\eta; q(t_n)]\right) d\eta}_{lte^{\text{SS}}_A}$$

$$+ \underbrace{\int_0^{\Delta t} B\left(q_B\left[\eta; q_A[\Delta t; q(t_n)]\right]\right) d\eta - \int_0^{\Delta t} B\left(q[\eta, q(t_n)]\right) d\eta}_{lte^{\text{SS}}_B}. \tag{12}$$

Here, the $lte^{\text{SS}}_A$ term is the same as $lte^{\text{PS}}_A$ in parallel splitting (see Eq. (9)) that includes only an isolation-induced error ($q_A \neq q$), while the $lte^{\text{SS}}_B$ term includes not only an isolation-induced error ($q_B \neq q$) but also an error resulting from the $B$ process being integrated from an input that has already been updated by the $A$ process, namely, $q_A[\Delta t; q(t_n)]$.

## 2.3 The leading-order error terms

The analysis in the previous subsections provides a qualitative understanding of the sources of errors associated with different coupling methods. In order to obtain a more quantitative assessment of the error magnitudes and identify possible cancelation of different error terms, Taylor series expansion is used to derive the leading-order error terms of the local truncation error. The gist of the method is to expand the integrals in Sect. 2.2 about $\Delta t = 0$, noting that

- $q_{X_i}(t)$ and $q(t)$ are different functions whose time derivatives are given by Eq. (1) and Eq. (3), respectively;

- The input state $q^{\text{input}}_{X_i}$ used for integrating the ODE of process $X_i$ might deviate from the exact solution at $t_n$ in a way that depends on $\Delta t$, and hence also need an expansion. For example, in the case of sequential splitting, $q^{\text{input}}_B = q_A[\Delta t; q(t_n)]$.

Appendix A explains in detail how the various integrals in Sect. 2.2 can be expanded to derive the leading-order error terms for the parallel and sequential splitting methods. Below, the key pieces of information obtained through these derivations are highlighted.

**Leading-order errors in parallel splitting**

The derivation detailed in Appendix A2 indicates that, when parallel splitting is used, the local truncation error attributable to the integration of process $A$ (i.e., the part marked with $lte_A^{\mathrm{PS}}$ in Eq. (9)) is

$$lte_A^{\mathrm{PS}} = \frac{(\Delta t)^2}{2}\left(-\frac{dA}{dq}B\right)\bigg|_{q=q(t_n)} + \mathcal{O}\left((\Delta t)^3\right). \tag{13}$$

From the derivation in Appendix A2, it can be seen that the leading-order error (the $(\Delta t)^2$ term) results from how the equation of $dq_A/dt$ lacks the $B$ term that appears in the equation of $dq/dt$. In other words, the leading-order error is caused by integrating the $A$ process in isolation without considering the influence of $B$. Similarly, the local truncation error attributable to the integration of process $B$ (i.e., the part marked with $lte_B^{\mathrm{PS}}$ in Eq. (9)) is

$$lte_B^{\mathrm{PS}} = \frac{(\Delta t)^2}{2}\left(-\frac{dB}{dq}A\right)\bigg|_{q=q(t_n)} + \mathcal{O}\left((\Delta t)^3\right). \tag{14}$$

The leading-order error term is caused by applying the $B$ process in isolation without considering the influence of $A$. The overall local truncation error for parallel splitting reads

$$\begin{aligned}
\mathcal{F}_{\Delta t}^{\mathrm{PS}}\left(q(t_n)\right) - q(t_{n+1}) &= lte_A^{\mathrm{PS}} + lte_B^{\mathrm{PS}} \\
&= \frac{(\Delta t)^2}{2}\left(-\frac{dA}{dq}B - \frac{dB}{dq}A\right)\bigg|_{q=q(t_n)} + \mathcal{O}\left((\Delta t)^3\right).
\end{aligned} \tag{15}$$

As mentioned earlier, the expression has symmetry between the processes $A$ and $B$, as the parallel splitting method treats the two processes in the same way.

**Leading-order errors in sequential splitting**

The derivation detailed in Appendix A3 shows that, when sequential splitting is used, the local truncation errors attributable to the integration of processes $A$ and $B$ are

$$lte_A^{\mathrm{SS}} = \frac{(\Delta t)^2}{2}\left(-\frac{dA}{dq}B\right)\bigg|_{q=q(t_n)} + \mathcal{O}\left((\Delta t)^3\right), \tag{16}$$

$$lte_B^{\mathrm{SS}} = \frac{(\Delta t)^2}{2}\left(+\frac{dB}{dq}A\right)\bigg|_{q=q(t_n)} + \mathcal{O}\left((\Delta t)^3\right), \tag{17}$$

respectively. Here, $lte_A^{\mathrm{SS}}$ has the same expression as in parallel splitting (see Eq. (16) versus Eq. (13)); $lte_B^{\mathrm{SS}}$ has the same form as in parallel splitting but a different sign, which results from the fact that the $B$ process is integrated with an input state already

updated by the $A$ process, and the input-induced error overcompensates the error caused by ignoring the influence of $A$ when integrating the $B$ process (see Appendix A3). The overall local truncation error for sequential splitting is

$$
\begin{aligned}
\mathcal{F}_{\Delta t}^{\mathrm{SS}}\Big(q(t_n)\Big) - q(t_{n+1}) &= lte_A^{\mathrm{SS}} + lte_B^{\mathrm{SS}} \\
&= \frac{(\Delta t)^2}{2}\left(-\frac{dA}{dq}B + \frac{dB}{dq}A\right)\bigg|_{q=q(t_n)} + \mathcal{O}\left((\Delta t)^3\right).
\end{aligned}
\tag{18}
$$

## 2.4 Framework Summary and Generalization

Leveraging the framework to derive and compare the splitting truncation errors of parallel and sequential splitting provides the following understanding that generalizes beyond the two-process problem. Note that a term containing $(dB/dq)\,A$ indicates an error caused by inaccurate accounts of the influence of process $A$ on process $B$. This can be seen from the following Taylor expansion:

$$
\begin{aligned}
\Delta t\left(\frac{dB}{dq}A\right)\bigg|_{q=q(t_n)} = &\, B\Big(q(t_n) + \Delta t A\big(q(t_n)\big)\Big) - B\big(q(t_n)\big) \\
&+ \mathcal{O}\left((\Delta t)^2\right).
\end{aligned}
$$

A negative sign in front of $(dB/dq)\,A$ suggests a lack or underestimation of the influence of $A$ on $B$, while a positive sign suggests an overestimation of that influence. Isolation-induced error in the form of integrating $B$ in isolation will lead to an underestimation of the influence of $A$ on $B$. If there are additional processes, integrating $B$ in isolation will lead to an underestimation of the influence of all other processes on $B$. Input-induced error in the form of using an input state updated by a full timestep worth of $A$ will lead to an overestimation of the influence of $A$ on $B$. If there are additional processes, using an input state updated by a full timestep worth of $A$ and other processes will lead to an overestimation of $A$ and those other processes on $B$. With this understanding, the framework is easily generalized to coupling methods that utilize different combinations of input states across numerous processes. The following section will demonstrate such a generalization to a three-process problem.

It is also worthwhile to note that the framework presented here can be revised to evaluate the overall truncation error in a temporally discretized system. Namely, one would replace $q_{X_i}[t - t_n; q_{X_i}^{\mathrm{Input}}]$ in Appendix A1 with the time integration scheme used for process $X_i$. For example, if forward Euler is used for integration of process $X_i$, one would use

$$
q_{X_i}[t - t_n; q_{X_i}^{\mathrm{input}}] \equiv q_{X_i}^{\mathrm{input}} + (t - t_n)X_i\big(q_{X_i}^{\mathrm{input}}\big).
$$

Such a revised framework would lead to results equivalent to those presented in, e.g., Ubbiali et al. (2021). That said, it is also worthwhile to note that such overall truncation error results can be more difficult to interpret than the results of the framework as presented here. Take, for example, how the overall truncation error for parallel splitting with forward Euler time integration of all processes is identical to the overall truncation for an unsplit forward Euler approach, which may seem to suggest that there is no splitting truncation error for the parallel splitting method in general. However, the overall truncation error expressions for parallel splitting and the unsplit approach will not be identical if backward Euler time integration is instead used, revealing that there is indeed splitting truncation error for the parallel splitting method.

## 3 The semi-discrete analysis framework applied to a three-process problem inspired by EAMv1

The semi-discrete error analysis framework presented in Sect. 2 is now used to analyze the dust life cycle problem in EAMv1. The dust mass budget analyses carried out using EAMv1's simulation output and presented in Sect. 3.1 of Part I (Wan et al., 2023) have revealed that at the global scale, the strongest sources and sinks of dust aerosols are (i) surface emissions, (ii) dry removal, and (iii) turbulent mixing and aerosol activation–resuspension. As such, we focus on these sources and sinks, ignore the many other aerosol-related processes in EAM, and consider a canonical three-process problem

$$\frac{dq(t)}{dt} = A\Big(q(t)\Big) + B\Big(q(t)\Big) + C\Big(q(t)\Big), \ t > 0, \ q(0) = q^{\text{IC}} \tag{19}$$

where $q$ is a dust mass mixing ratio, $A$ represents the emissions, $B$ represents dry removal, and $C$ corresponds to turbulent mixing. As in Sect. 2, denote a discrete time step $\Delta t$ and discrete time points $t_{n+1} = t_n + \Delta t$. Denote the numerical solution at time $t_{n+1}$ as $q^{n+1}$ and the exact solution it approximates as $q(t_{n+1})$. Fig. 2 describes two schemes for obtaining $q^{n+1}$ from $q^n$, corresponding to the original and revised process coupling schemes in EAMv1 discussed in the Part I paper.

Consider three single-process ODEs in the form of Eq. (3) where $X_i = A, B$, or $C$, namely,

$$\frac{dq_A(t)}{dt} = A\Big(q(t)\Big), \ t > t_n, \ q_A(t_n) = q_A^{\text{input}},$$

$$\frac{dq_B(t)}{dt} = B\Big(q(t)\Big), \ t > t_n, \ q_B(t_n) = q_B^{\text{input}},$$

$$\frac{dq_C(t)}{dt} = C\Big(q(t)\Big), \ t > t_n, \ q_C(t_n) = q_C^{\text{input}},$$

The exact solutions are denoted using the notation defined in Eq. (4), namely,

$$q_A[t - t_n; q_A^{\text{input}}] \equiv q_A^{\text{input}} + \int_{t_n}^{t} A\Big(q_A(\eta)\Big) d\eta$$

$$= q_A^{\text{input}} + \int_{0}^{t-t_n} A\Big(q_A[\tilde{\eta}; q_A^{\text{input}}]\Big) d\tilde{\eta},$$

$$q_B[t - t_n; q_B^{\text{input}}] \equiv q_B^{\text{input}} + \int_{t_n}^{t} B\Big(q_B(\eta)\Big) d\eta$$

$$= q_B^{\text{input}} + \int_{0}^{t-t_n} B\Big(q_B[\tilde{\eta}; q_B^{\text{input}}]\Big) d\tilde{\eta},$$

$$q_C[t - t_n; q_C^{\text{input}}] \equiv q_C^{\text{input}} + \int_{t_n}^{t} C\Big(q_C(\eta)\Big) d\eta$$

$$= q_C^{\text{input}} + \int_{0}^{t-t_n} C\Big(q_C[\tilde{\eta}; q_C^{\text{input}}]\Big) d\tilde{\eta}.$$

**(a1) Original scheme in EAMv1, flowchart description**

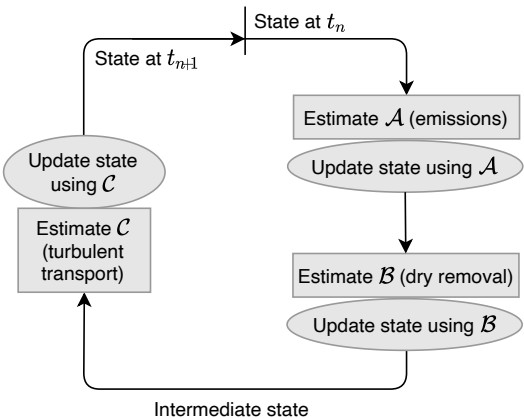

**(a2) Revised scheme, flowchart description**

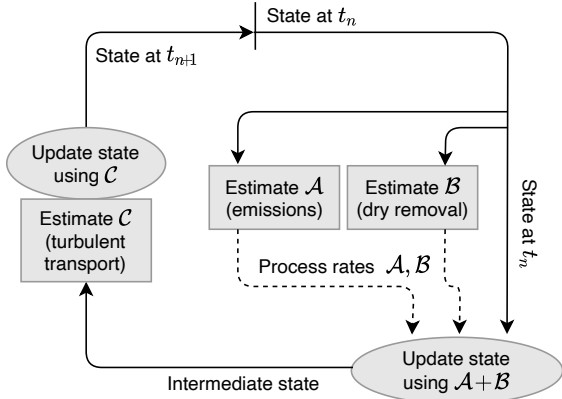

**(b1) Original scheme in EAMv1, pseudo code description**

$$q^n \xrightarrow{\text{apply } A \text{ process for duration } \Delta t} q_A^{n+1}$$

$$q_A^{n+1} \xrightarrow{\text{apply } B \text{ process for duration } \Delta t} q_B^{n+1}$$

$$q_B^{n+1} \xrightarrow{\text{apply } C \text{ process for duration } \Delta t} q_C^{n+1}$$

Let $q^{n+1} = q_C^{n+1}$.

**(b2) Revised scheme, pseudo code description**

$$q^n \xrightarrow{\text{apply } A \text{ process for duration } \Delta t} q_A^{n+1}; \text{ let } A^* = \frac{q_A^{n+1} - q^n}{\Delta t}$$

$$q^n \xrightarrow{\text{apply } B \text{ process for duration } \Delta t} q_B^{n+1}; \text{ let } B^* = \frac{q_B^{n+1} - q^n}{\Delta t}$$

$$q^n + \Delta t \left( A^* + B^* \right) \xrightarrow{\text{apply } C \text{ process for duration } \Delta t} q_C^{n+1}$$

Let $q^{n+1} = q_C^{n+1}$.

**(c1) Original scheme in EAMv1, ODE description**

1. Solve for $q_A(t_{n+1})$ given $\dfrac{dq_A}{dt} = A(q_A)$ with $q_A(t_n) = q^n$.

2. Solve for $q_B(t_{n+1})$ given $\dfrac{dq_B}{dt} = B(q_B)$ with $q_B(t_n) = q_A^{n+1}$.

3. Solve for $q_C(t_{n+1})$ given $\dfrac{dq_C}{dt} = C(q_C)$ with $q_C^n = q_B^{n+1}$.

4. Let $q^{n+1} = q_C^{n+1}$.

**(c2) Revised scheme, ODE description**

1. Solve for $q_A(t_{n+1})$ given $\dfrac{dq_A}{dt} = A(q_A)$ with $q_A(t_n) = q^n$.

2. Solve for $q_B(t_{n+1})$ given $\dfrac{dq_B}{dt} = B(q_B)$ with $q_B(t_n) = q^n$.

3. Let $A^* = \dfrac{q_A(t_{n+1}) - q^n}{\Delta t}$ and $B^* = \dfrac{q_B(t_{n+1}) - q^n}{\Delta t}$.

4. Solve for $q_C(t_{n+1})$ given $\dfrac{dq_C}{dt} = C(q_C)$ with $q_C(t_n) = q^n + \Delta t(A^* + B^*)$.

5. Let $q^{n+1} = q_C^{n+1}$.

**Figure 2.** Three different descriptions of two process coupling schemes for the three-process problem defined in Sect. 3. The scheme depicted in the left column corresponds to the original scheme used in EAMv1 for the coupling of aerosol emissions, dry removal, and the parameterization of turbulent transport and aerosol activation-resuspension. The scheme depicted in the right column corresponds to the revised scheme proposed and evaluated in the companion paper (Part I). We note that these descriptions are simplified versions of the coupling implemented in EAMv1. Here, we focus only on the three strongest sources and sinks of the global mean dust budget presented in Sect. 3 of the companion paper, while the many other processes in EAMv1 (see Fig. 1 in the companion paper) are omitted.

### 3.1 Process coupling schemes

The original EAMv1 uses sequential splitting for all three processes (see the left column in Fig. 2), which can be represented by the mapping

$$q^{n+1} = \mathcal{F}^{\text{Ori}}_{\Delta t}(q^n) \equiv q_C\Big[\Delta t; q_B\big[\Delta t; q_A[\Delta t; q^n]\big]\Big],\tag{20}$$

or equivalently,

$$q^{n+1} = q_C\Big[\Delta t; q^n + \big(q_A[\Delta t; q^n] - q^n\big)$$

$$+ \big(q_B\big[\Delta t; q_A[\Delta t; q^n]\big] - q_A[\Delta t; q^n]\big)\Big],\tag{21}$$

Defining

$$A^* \equiv \big(q_A[\Delta t; q^n] - q^n\big)/\Delta t,\tag{22}$$

$$B^* \equiv \big(q_B[\Delta t; q^n] - q^n\big)/\Delta t,\tag{23}$$

the revised coupling scheme depicted in the right column in Fig. 2 can be represented by the mapping

$$q^{n+1} = \mathcal{F}^{\text{Rev}}_{\Delta t}(q^n) \equiv q_C\Big[\Delta t; q^n + \Delta t\big(A^* + B^*\big)\Big]\tag{}$$

or equivalently,

$$q^{n+1} = q_C\Big[\Delta t; q^n + \big(q_A[\Delta t; q^n] - q^n\big)$$

$$+ \big(q_B[\Delta t; q^n] - q^n\big)\Big].\tag{24}$$

### 3.2 Leading-order error terms

The original and revised coupling schemes described in Eq. (20) and Eq. (24) can be viewed as different combinations of the two-process sequential and parallel splitting schemes discussed in Sect. 2. Based on the discussions in that section, and keeping in mind the focus here is the local truncation error, one can make the following reasoning about the original coupling scheme in EAMv1:

- For process $A$, since the solution procedure starts from the exact solution at $t_n$ and integrates the $A$ term in isolation, one expects to get order $(\Delta t)^2$ errors caused by performing time integration without considering the impacts of $B$ and $C$ on the $A$ process. The coefficients in front of $(dA/dq)\,B$ and $(dA/dq)\,C$ are expected to be $-(\Delta t)^2/2$. In other words, the splitting truncation error associated with the $A$ process is expected to be

$$lte^{\text{Ori}}_A = \frac{(\Delta t)^2}{2}\left[\frac{dA}{dq}(-B - C)\right]\bigg|_{q=q(t_n)} + \mathcal{O}\big((\Delta t)^3\big).\tag{25}$$

- For process $B$, since the solution procedure starts from a mixing ratio updated by $A$ and ignores the $A$ and $C$ terms on the RHS of the original ODE, one expects there to be two error terms caused by integrating $B$ in isolation and an error associated with the input state. The input-induced error is expected to overcompensate the error caused by ignoring the impact of $A$ on the $B$ process. Hence the splitting truncation error associated with the $B$ process is expected to have the form

$$lte_B^{\text{Ori}} = \frac{(\Delta t)^2}{2} \left[ \frac{dB}{dq}(+A-C) \right] \Bigg|_{q=q(t_n)} + \mathcal{O}\left((\Delta t)^3\right) \tag{26}$$

- For process $C$, the solution procedure starts from a mixing ratio updated by both $A$ and $B$, and the $C$ term is integrated in isolation. Therefore, one expects to have two input-induced errors overcompensating two isolation-induced errors, giving a splitting truncation error in the form of

$$lte_C^{\text{Ori}} = \frac{(\Delta t)^2}{2} \left[ \frac{dC}{dq}(+A+B) \right] \Bigg|_{q=q(t_n)} + \mathcal{O}\left((\Delta t)^3\right). \tag{27}$$

The overall local truncation error in the original coupling is expected to be

$$\mathcal{F}_{\Delta t}^{\text{Ori}}\left(q(t_n)\right) - q(t_{n+1}) = lte_A^{\text{Ori}} + lte_B^{\text{Ori}} + lte_C^{\text{Ori}}$$
$$= \frac{(\Delta t)^2}{2} \left[ \frac{dA}{dq}(-B-C) + \frac{dB}{dq}(A-C) + \frac{dC}{dq}(A+B) \right] \Bigg|_{q=q(t_n)}$$
$$+ \mathcal{O}\left((\Delta t)^3\right). \tag{28}$$

The revised coupling scheme differs from the original scheme only in the input state for the integration of the $B$ process, see Eq. (24) versus Eq. (21). Therefore, one expects the splitting truncation errors associated with the other two processes, $A$ and $C$, to be the same as in the original scheme, and that the splitting truncation error associated with the $B$ process to have a minus sign instead of + for the $(dB/dq)A$ term (i.e., no input-induced error, only the isolation-induced error). In other words,

$$lte_A^{\text{Rev}} = \frac{(\Delta t)^2}{2} \left[ \frac{dA}{dq}(-B-C) \right] \Bigg|_{q=q(t_n)} + \mathcal{O}\left((\Delta t)^3\right), \tag{29}$$

$$lte_B^{\text{Rev}} = \frac{(\Delta t)^2}{2} \left[ \frac{dB}{dq}(-A-C) \right] \Bigg|_{q=q(t_n)} + \mathcal{O}\left((\Delta t)^3\right), \tag{30}$$

$$lte_C^{\text{Rev}} = \frac{(\Delta t)^2}{2} \left[ \frac{dC}{dq}(+A+B) \right] \Bigg|_{q=q(t_n)} + \mathcal{O}\left((\Delta t)^3\right), \tag{31}$$

and the overall local truncation error is expected to be

$$\mathcal{F}_{\Delta t}^{\text{Rev}}\left(q(t_n)\right) - q(t_{n+1}) = lte_A^{\text{Rev}} + lte_B^{\text{Rev}} + lte_C^{\text{Rev}}$$
$$= \frac{(\Delta t)^2}{2} \left[ \frac{dA}{dq}(-B-C) + \frac{dB}{dq}(-A-C) + \frac{dC}{dq}(A+B) \right] \Bigg|_{q=q(t_n)}$$
$$+ \mathcal{O}\left((\Delta t)^3\right). \tag{32}$$

All of the error expressions in Eq. (25)–Eq. (32) are confirmed by the step-by-step derivations presented in Appendix B. This agreement demonstrates how the two-process splitting truncation error results can be leveraged as building blocks to derive splitting truncation errors for multi-process problems using logical reasoning instead of Taylor series expansions. In other words, the derivation of splitting truncation error for multi-process problems does not always have to be done in the step-by-365 step manner in Appendix B. The logical reasoning approach not only can facilitate rapid development of alternative coupling schemes but also makes the framework more accessible to model developers on the physics side who might find the lengthy calculus derivations too tedious or daunting.

### 3.3 Characteristics of the leading-order error terms inferred from EAMv1 results

Recall that the three RHS terms in the three-process ODE discussed above are meant to represent surface emissions ($A$),
dry removal ($B$), and turbulent mixing ($C$), respectively, of dust aerosols in EAMv1. The parameterization descriptions and EAMv1 simulations presented in Sects. 2 and 3 of the Part I paper can be used to infer several features of the leading-order error terms listed above in Eq. (3.2). The dust budget analyses shown in Sect. 3 of Part I indicate that the dominant sources and sinks are found in the lowest model layer in the dust source regions, where $A$ (emission) is a source, $B$ (dry removal) is typically a sink, and $C$ (turbulent mixing) is typically a sink (i.e., $A > 0$, $B < 0$, and $C < 0$). Given the same air density and
deposition velocity, the downward dry removal flux at the Earth's surface is proportional to the mean dust mixing ratio of the layer (see Eq. 1 in Part I). This means one can expect

$$\frac{dB}{dq} < 0 \tag{33}$$

to be true in typical cases. Equation (33) is confirmed by the scatter plot in Fig. 3 where the dry removal rate $B$ is plotted against dust mixing ratio $q$ using 90 days of 6-hourly output in dust source regions simulated with the original EAMv1. It then
follows that the local truncation error associated with the $B$ process in EAMv1's original process coupling (see Eq. (26)) can be written as

$$lte_B^{\mathrm{Ori}} = -\frac{(\Delta t)^2}{2} \left[ \left| \frac{dB}{dq} \right| \left( |A| + |C| \right) \right] \Bigg|_{q=q(t_n)} + \mathcal{O}\left( (\Delta t)^3 \right), \tag{34}$$

and the local truncation error of the $B$ process in the revised coupling scheme (see Eq. (30)) can be written as

$$lte_B^{\mathrm{Rev}} = \frac{(\Delta t)^2}{2} \left[ \left| \frac{dB}{dq} \right| \left( |A| - |C| \right) \right] \Bigg|_{q=q(t_n)} + \mathcal{O}\left( (\Delta t)^3 \right). \tag{35}$$

Because $\left| |A| - |C| \right| \leq |A| + |C|$, with equality only when $A$ or $C$ is zero, it is expected that the magnitude of the leading-order local truncation error associated with process $B$ to be smaller in the revised coupling than in the original scheme, i.e.,

$$\left| lte_B^{\mathrm{Rev}} \right| \lesssim \left| lte_B^{\mathrm{Ori}} \right|. \tag{36}$$

This result, combined with the fact that the local truncation errors associated with the other two processes ($A$ and $C$) have the same expressions in the original and revised coupling schemes, provides a justification for adopting the revised scheme in
EAMv1.

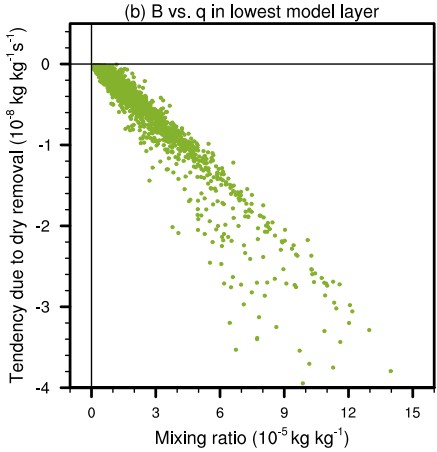

**Figure 3.** Dust aerosol dry removal rate (y-axis) plotted against dust aerosol mixing ratio (x-axis) in the lowest model layer in dust sources regions simulated by the original EAMv1 using a vertical grid with 72 layers. The data used in the figure included 90 days of 6-hourly instantaneous output.

To see that Eq. (36) holds in practice, it is useful to first note that the main leading-order difference between $lte_B^{\mathrm{Ori}}$ in Eq. (26) and $lte_B^{\mathrm{Rev}}$ in Eq. (30) is the term $A - C$ versus $(-A - C)$, respectively, evaluated at $q = q(t_n)$. While the values of $A\big(q(t_n)\big)$ and $C\big(q(t_n)\big)$ are not feasible to obtain in practice, as $q(t_n)$ is the exact solution, the approximate $A$ and $C$ values calculated and used in EAMv1 simulations are relatively straightforward to obtain using the online diagnostic tool of Wan et al. (2022), as was done in Part I. Figure 4 shows annual mean values of the computed $(A - C)$ and $(-A - C)$ in dust source regions in North Africa, using both the original and revised coupling methods. When the original coupling method is used, the magnitude of $(A - C)$ ranges from being slightly to substantially larger than the magnitude of $(-A - C)$. When the revised coupling is used, the magnitude of $(A - C)$ is substantially larger everywhere than that of $(-A - C)$. Both results support that the leading-order term in $lte_B^{\mathrm{Rev}}$ is smaller in magnitude than that of the leading-order term in $lte_B^{\mathrm{Ori}}$, i.e., Eq. (36).

It is also worth noting the leading-order term in $lte_B^{\mathrm{Ori}}$ is negative, by Eq. (34), and because $B$ itself is negative, the negative leading-order term in the truncation error indicates an overestimation of the $B$ process at each timestep of the original coupling method. The study by Feng et al. (2022) has pointed out that dust dry removal in EAMv1 is generally overestimated in dust source regions, and thus the reduction of $|lte_B|$ shown in Eq. (36) is consistent with the significantly weaker dry removal seen in the Part I work (Wan et al., 2023) when the revised method is used instead of the original method in EAMv1. While the local truncation error caused by process splitting is not the only source of error in global simulations (other error sources include, e.g., propagated splitting error in Eq. (2), temporal and spatial discretization errors in dry removal and other aerosol processes, model formulation error, parameter uncertainty, etc.), the theoretical analysis here and the global simulations in Feng et al. (2022) and Wan et al. (2023) suggest $lte_B^{\mathrm{Ori}}$ is likely an important contributor to the overly strong dust dry removal in the original EAMv1.

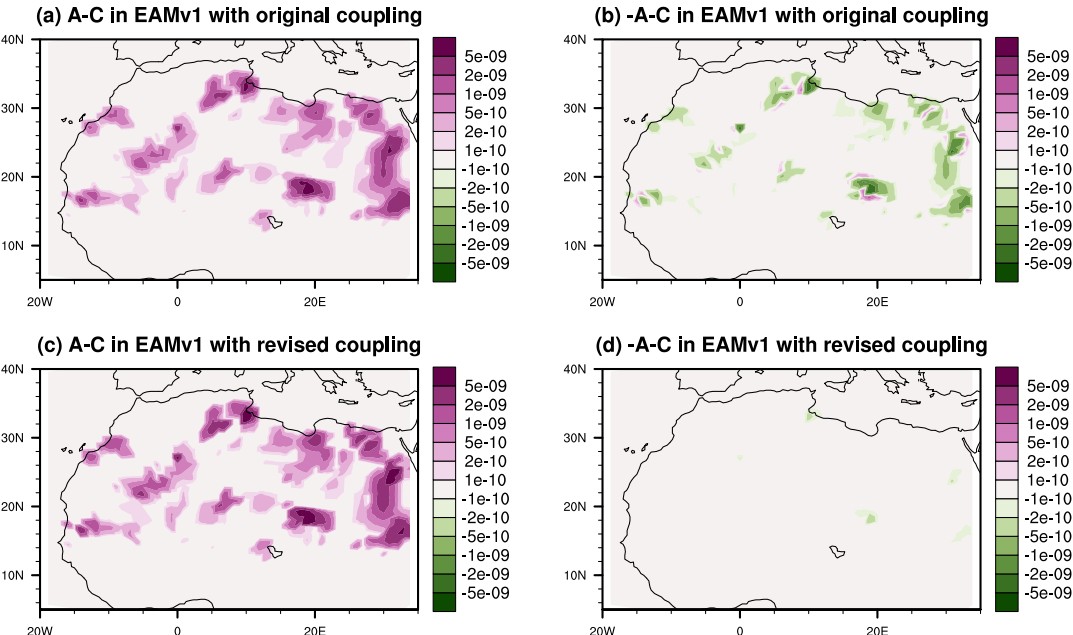

**Figure 4.** Comparison of key terms in the splitting truncation error associated with dry removal (process $B$) using 10-year mean interstitial dust mass mixing ratio process rates (unit: kg kg$^{-1}$ s$^{-1}$) caused by emissions (process $A$) and dry removal (process $B$) in the lowest model layer in EAMv1 simulations using the original coupling scheme (upper row) and the revised scheme (lower row).

## 4 Summary and conclusions

A semi-discrete error analysis framework was introduced for assessing splitting error of process coupling methods. By assuming the time integration of each individual process is exact, the framework identified two general sources of splitting error. The first is denoted isolation-induced error and is from the integration, exact or otherwise, of a process without the influence of other processes (i.e., in isolation). The second is denoted input-induced error and is from starting the single-process integration from an input that deviates from the full solution of the multi-process equation. The corresponding splitting truncation error terms from those two sources were derived for a generic two-process problem for two common coupling methods: parallel splitting and sequential splitting. The parallel splitting method results in isolation-induced error from both processes. The combination of isolation-induced errors leads to an underestimation of the influence of the one process on the other. The sequential splitting method results in isolation-induced error from the first process that is integrated in a timestep and in both types of error for the second process. The combination of the two types of errors leads to an underestimation of the influence of the second process on the first, as in parallel splitting, but the input-induced error from the second process is shown to overcompensate the isolation-induced error from the first process. Thus, sequential splitting results in an overestimate of the influence of the first process on the second process.

A three-process problem and two coupling schemes were analyzed that corresponded to the coupling of dust emissions, dry
removal, and turbulence mixing in the original EAMv1 and the revised coupling scheme proposed in the Part I paper. The
semi-discrete analysis revealed that the original and revised coupling schemes have the same forms of leading-order error for
the emissions and turbulent mixing, while the magnitude of the local truncation error in dry removal is smaller in the revised
scheme. Assuming it is useful, especially in the long run, to address different error sources in EAM separately, this result
provides a justification to adopt the revised coupling in EAMv1. The analysis also revealed that the local truncation error of the
430 dry removal process in the original EAMv1 corresponds to an overestimation of the dry removal rate. This result, combined
with the EAMv1 results presented in Feng et al. (2022) and the Part I paper, suggests that the sequential splitting of emissions,
dry removal, and turbulent mixing is likely an important contributor to the overestimated dry removal in dust source regions in
the original EAMv1.

While the error analysis framework is presented in the context of discussions on the dust life cycle in EAMv1, using the
435 framework as done in this work is much more general. For one, such a framework can be used to analyze coupling methods
beyond the two schemes discussed here and in Part I as well as numerical coupling problems involving more than three
processes. Additionally, because many applications rely on low order coupling methods such as those discussed here, this
paper shows how such a framework can be used to inform choices of coupling approaches in areas beyond atmospheric climate,
including hydrology, fusion, reactive flow modeling, and many others. As such, the authors plan to use the framework to help
further reduce splitting errors in EAM and other applications.

*Code and data availability.* The EAMv1 source code used in this study can be found on Zenodo as record 7995850 (Wan, 2023). The decadal
mean and instantaneous model output used for figures in this paper can be found on NERSC (Wan and Zhang, 2023).

**Appendix A: Derivation of the leading-order error terms in a generic two-process ODE**

This section details the step-by-step derivation of the leading-order local truncation error terms caused by applying the parallel
splitting and sequential splitting methods to solving the two-process problem defined in Eq. (7). The starting point is the local
truncation error terms $lte_A^{\mathrm{SS}}$ and $lte_B^{\mathrm{SS}}$ in Eq. (12) and $lte_A^{\mathrm{PS}}$ and $lte_B^{\mathrm{PS}}$ in Eq. (9).

**A1   Taylor expansion of an integral**

The Taylor expansion of an integral is a key element in deriving the leading-order terms. As such, it is formalized herein. Recall
from Eq. (5) that $q[\eta; \phi]$ is the exact solution of the multi-process problem Eq. (1) evolved from input state $\phi$ for time $\eta$. For a
450 function $f(\delta)$ defined as

$$f(\delta) = \int_0^\delta F\Big(q[\eta; \phi]\Big)\, d\eta,$$

the first and second derivatives are

$$f'(\delta) = F\Big(q[\delta;\phi]\Big),$$

$$f''(\delta) = \frac{dF}{dq}\big(q[\delta;\phi]\big)\frac{dq}{dt}[\delta;\phi],$$

respectively, and thus

$$f(0) = 0,$$

$$f'(0) = F\Big(q[0;\phi]\Big) = F(\phi),$$

$$f''(0) = \frac{dF}{dq}\big(q[0,\phi]\big)\frac{dq}{dt}[0;\phi] = \frac{dF}{dq}(\phi)\frac{dq}{dt}[0;\phi].$$

Note that Eq. (6) was used to simplify the expression. The Taylor expansion of $f(\Delta t)$ about $\Delta t = 0$ is now given as

$$
\begin{aligned}
f(\Delta t) =& f(0) + \Delta t\, f'(0) + \frac{(\Delta t)^2}{2}f''(0) + \mathcal{O}\big((\Delta t)^3\big) \\
=& 0 + \Delta t\, F(\phi) + \frac{(\Delta t)^2}{2}\frac{dF}{dq}(\phi)\frac{dq}{dt}[0;\phi] \\
& + \mathcal{O}\big((\Delta t)^3\big).
\end{aligned}
\tag{A1}
$$

Note that Eq. (A1) can be used to show both that

$$\int_0^{\Delta t} A\big(q[\eta;q(t_n)]\big)\,d\eta = \Delta t A\big(q(t_n)\big) + \frac{(\Delta t)^2}{2}\left.\left(\frac{dA}{dq}(A+B)\right)\right|_{q=q(t_n)} + \mathcal{O}\big((\Delta t)^3\big) \tag{A2}$$

and

$$\int_0^{\Delta t} B\big(q[\eta;q(t_n)]\big)\,d\eta = \Delta t B\big(q(t_n)\big) + \frac{(\Delta t)^2}{2}\left.\left(\frac{dB}{dq}(A+B)\right)\right|_{q=q(t_n)} + \mathcal{O}\big((\Delta t)^3\big) \tag{A3}$$

## A2 Parallel splitting

Recall the local truncation error term for parallel splitting in Eq. (9):

$$lte_A^{\text{PS}} = \int_0^{\Delta t} A\big(q_A[\eta;q(t_n)]\big)\,d\eta - \int_0^{\Delta t} A\big(q[\eta;q(t_n)]\big)d\eta. \tag{A4}$$

The second integral is expanded using Eq. (A2). For the first integral, use Eq. (A1), with $F = A$, $q = q_A$, and $\phi = q(t_n)$ so that $F\big(q[\eta;\phi]\big) = A\big(q_A[\eta;q(t_n)]\big)$, to find

$$\int_0^{\Delta t} A\big(q_A[\eta;q(t_n)]\big)\,d\eta = \Delta t A\big(q(t_n)\big) + \frac{(\Delta t)^2}{2}\frac{dA}{dq}\big(q(t_n)\big)\frac{dq_A}{dt}[0;q(t_n)] + \mathcal{O}\big((\Delta t)^3\big)$$

which can be simplified using Eq. (3) to get

$$\int_0^{\Delta t} A\big(q_A[\eta; q(t_n)]\big)\, d\eta = \Delta t A\big(q(t_n)\big) + \frac{(\Delta t)^2}{2}\left(\frac{dA}{dq}A\right)\bigg|_{q=q(t_n)} + \mathcal{O}((\Delta t)^3).$$

The expansions of the integrals in Eq. (A4) are now combined to find

$$lte_A^{\text{PS}} = -\frac{(\Delta t)^2}{2}\left(\frac{dA}{dq}B\right)\bigg|_{q=q(t_n)} + \mathcal{O}((\Delta t)^3),$$

as shown in Eq. (13). Recall the other local truncation error term from Eq. (9):

$$lte_B^{\text{PS}} = \int_0^{\Delta t} B\big(q_B[\eta; q(t_n)]\big)\, d\eta - \int_0^{\Delta t} B\big(q[\eta; q(t_n)]\big)\, d\eta. \tag{A5}$$

The second integral is expanded using Eq. (A3). For the first integral, use Eq. (A1), with $F = B$, $q = q_B$, and $\phi = q(t_n)$ so that $F\big(q[\eta; \phi]\big) = B\big(q_B[\eta; q(t_n)]\big)$, to find

$$\begin{aligned}
\int_0^{\Delta t} B\big(q_B[\eta; q(t_n)]\big)\, d\eta &= \Delta t\, B\big(q(t_n)\big) \\
&+ \frac{(\Delta t)^2}{2}\frac{dB}{dq}\big(q(t_n)\big)\frac{dq_B}{dt}[0; q(t_n)] \\
&+ \mathcal{O}\big((\Delta t)^3\big),
\end{aligned} \tag{A6}$$

which can be simplified using Eq. (3) to get

$$\begin{aligned}
\int_0^{\Delta t} B\big(q_B[\eta; q^n]\big)\, d\eta &= \Delta t\, B\big(q(t_n)\big) \\
&+ \frac{(\Delta t)^2}{2}\left(\frac{dB}{dq}B\right)\bigg|_{q=q(t_n)} + \mathcal{O}\big((\Delta t)^3\big).
\end{aligned}$$

The expansions of the integrals in Eq. (A5) are now combined to find

$$lte_B^{\text{PS}} = -\frac{(\Delta t)^2}{2}\left(\frac{dB}{dq}A\right)\bigg|_{q=q(t_n)} + \mathcal{O}((\Delta t)^3)$$

as shown in Eq. (14).

## A3  Sequential splitting

Recall the local truncation error term for sequential splitting in Eq. (12):

$$lte_A^{\text{SS}} = \int_0^{\Delta t} A\big(q_A[\eta; q(t_n)]\big)\, d\eta - \int_0^{\Delta t} A\big(q[\eta; q(t_n)]\big)\, d\eta.$$

Note that $lte_A^{SS}$ is equivalent to $lte_A^{PS}$, which has already been derived in Sect. A2 and is equivalent to Eq. (16). Recall the other local truncation error term from Eq. (12):

$$lte_B^{SS} = \int_0^{\Delta t} B\big(q_B\left[\eta; q_A[\Delta t; q(t_n)]\right]\big)\, d\eta - \int_0^{\Delta t} B\big(q[\eta; q(t_n)]\big)\, d\eta \tag{A7}$$

The second integral is expanded using Eq. (A3). For the first integral, use Eq. (A1), with $F = B$, $q = q_B$, and $\phi = q_A[\Delta t; q(t_n)]$ so that $F\Big(q[\eta;\phi]\Big) = B\Big(q_B[\eta; q_A[\Delta t; q(t_n)]]\Big)$, to find

$$\int_0^{\Delta t} B\left( q_B\Big[\eta; q_A[\Delta t; q(t_n)]\Big]\right) d\eta = \Delta t\, B\left( q_A[\Delta t; q(t_n)]\Big]\right)$$
$$+ \frac{(\Delta t)^2}{2} \frac{dB}{dq}\big(q_A[\Delta t; q(t_n)]\big) \frac{dq_B}{dt}\big[0, q_A[\Delta t; q(t_n)]\big]$$
$$+ \mathcal{O}\big((\Delta t)^3\big)$$

which can be simplified using Eq. (3) to get

$$\int_0^{\Delta t} B\left( q_B\Big[\eta; q_A[\Delta t; q(t_n)]\Big]\right) d\eta = \Delta t\, B\left( q_A[\Delta t; q(t_n)]\right)$$
$$+ \frac{(\Delta t)^2}{2} \left( \frac{dB}{dq} B\right)\Bigg|_{q = q_A[\Delta t; q(t_n)]} + \mathcal{O}\big((\Delta t)^3\big). \tag{A8}$$

To continue the expansion, use

$$q_A[\Delta t; q(t_n)] = q_A[0; q(t_n)] + \Delta t \frac{dq_A}{dt}[0; q(t_n)] + \mathcal{O}\big((\Delta t)^2\big)$$
$$= q(t_n) + \Delta t A\big(q(t_n)\big) + \mathcal{O}\big((\Delta t)^2\big) \tag{A9}$$

to get

$$\Delta t\, B\Big(q_A[\Delta t; q(t_n)]\Big)$$
$$= \Delta t\, B\big(q(t_n)\big) + (\Delta t)^2 \left( \frac{dB}{dq} A\right)\Bigg|_{q = q(t_n)} + \mathcal{O}\big((\Delta t)^3\big)$$

and

$$\frac{(\Delta t)^2}{2} \left( \frac{dB}{dq} B\right)\Bigg|_{q = q_A[\Delta t; q^n]}$$
$$= \frac{(\Delta t)^2}{2} \left( \frac{dB}{dq} B\right)\Bigg|_{q = q(t_n)} + \mathcal{O}\big((\Delta t)^3\big)$$

This allows for the simplification of Eq. (A8) to

$$
\int_0^{\Delta t} B\Big( q_B\big[\eta; q_A[\Delta t; q(t_n)]\big] \Big)\, d\eta
$$

$$
= \Delta t\, B\big(q(t_n)\big) + (\Delta t)^2 \left( \frac{dB}{dq} A \right)\bigg|_{q=q(t_n)} + \frac{(\Delta t)^2}{2} \left( \frac{dB}{dq} B \right)\bigg|_{q=q(t_n)}
$$

$$
+ \mathcal{O}\big((\Delta t)^3\big)
$$

(A10)

The expansions of the integrals in Eq. (A7) are now combined to find

$$
lte_B^{\mathrm{SS}} = \frac{(\Delta t)^2}{2} \left( \frac{dB}{dq} A \right)\bigg|_{q=q(t_n)} + \mathcal{O}\big((\Delta t)^3\big),
$$

as shown in Eq. (17). It is worth noting that compared to the expansion of Eq. (A6), the expansion of Eq. (A8) has an extra term $(\Delta t)^2 \left( \frac{dB}{dq} A \right)\big|_{q=q(t_n)}$ that results from the sequential splitting method using $q_A[\Delta t; q(t_n)]$ (the value of $q$ already updated by process $A$) as the input when integrating the $dq_B/dt$ equation. This leads to the sign difference between $lte_B^{\mathrm{SS}}$ and $lte_B^{\mathrm{PS}}$ that can be traced to the fact that the input used when integrating process $B$, i.e., $q_A[\Delta t; q(t_n)]$, results in a leading-order error in the solution that overcompensates the leading-order term caused by integrating the equation of $dq_B/dt$ without an $A$ term on the right-hand side.

## Appendix B: Derivation of the leading-order error terms in a three-process ODE

This section details the derivation of the leading-order local truncation error terms caused by the original splitting and revised splitting methods to solving the three-process problem defined in Eq. (19). As it will be useful herein, start by using Eq. (A1) to show that

$$
\int_0^{\Delta t} A\big(q[\eta; q(t_n)]\big)\, d\eta = \Delta t A\big(q(t_n)\big) + \frac{(\Delta t)^2}{2} \left( \frac{dA}{dq}(A+B+C) \right)\bigg|_{q=q(t_n)} + \mathcal{O}\big((\Delta t)^3\big),
$$

(B1)

$$
\int_0^{\Delta t} B\big(q[\eta; q(t_n)]\big)\, d\eta = \Delta t B\big(q(t_n)\big) + \frac{(\Delta t)^2}{2} \left( \frac{dB}{dq}(A+B+C) \right)\bigg|_{q=q(t_n)} + \mathcal{O}\big((\Delta t)^3\big),
$$

(B2)

and

$$
\int_0^{\Delta t} C\big(q[\eta; q(t_n)]\big)\, d\eta = \Delta t C\big(q(t_n)\big) + \frac{(\Delta t)^2}{2} \left( \frac{dC}{dq}(A+B+C) \right)\bigg|_{q=q(t_n)} + \mathcal{O}\big((\Delta t)^3\big).
$$

(B3)

## B1 Revised splitting

The mapping in the revised splitting described in Eq. (24) can be written as

$$\mathcal{F}^{\text{Rev}}_{\Delta t}\big(q(t_n)\big) = q_C\big[\Delta t; q(t_n) + \Delta t(A^* + B^*)\big] = q(t_n) + \Delta t(A^* + B^*) + \int_0^{\Delta t} C\big(q_C[\eta; q(t_n) + \Delta t(A^* + B^*)]\big)\, d\eta$$

$$= q(t_n) + \int_0^{\Delta t} A\big(q_A[\eta; q(t_n)]\big)\, d\eta + \int_0^{\Delta t} B\big(q_B[\eta; q(t_n)]\big)\, d\eta + \int_0^{\Delta t} C\big(q_C[\eta; q(t_n) + \Delta t(A^* + B^*)]\big)\, d\eta.$$

Thus, the local truncation error is expressed as

$$\mathcal{F}^{\text{Rev}}_{\Delta t}\big(q(t_n)\big) - q(t_{n+1}) = \underbrace{\int_0^{\Delta t} A\big(q_A[\eta; q(t_n)]\big)\, d\eta - \int_0^{\Delta t} A\big(q[\eta; q(t_n)]\big)\, d\eta}_{lte^{\text{Rev}}_A} + \underbrace{\int_0^{\Delta t} B\big(q_B[\eta; q(t_n)]\big)\, d\eta - \int_0^{\Delta t} B\big(q[\eta; q(t_n)]\big)\, d\eta}_{lte^{\text{Rev}}_B}$$

$$+ \underbrace{\int_0^{\Delta t} C\big(q_C[\eta; q(t_n) + \Delta t(A^* + B^*)]\big)\, d\eta - \int_0^{\Delta t} C\big(q[\eta; q(t_n)]\big)\, d\eta}_{lte^{\text{Rev}}_C}$$

The leading order terms in each of $lte^{\text{Rev}}_A$, $lte^{\text{Rev}}_B$, and $lte^{\text{Rev}}_C$ will now be derived in a manner similar to their two-process counterparts in Sect. A. For $lte^{\text{Rev}}_A$, the second integral is expanded using Eq. (B1). For the first integral in $lte^{\text{Rev}}_A$, use Eq. (A1), with $F = A$, $q = q_A$, and $\phi = q(t_n)$ so that $F\big(q[\eta; \phi]\big) = A\big(q_A[\eta; \phi]\big)$, to find

$$\int_0^{\Delta t} A\big(q_A[\eta; q(t_n)]\big)\, d\eta = \Delta t A\big(q(t_n)\big) + \frac{(\Delta t)^2}{2} \frac{dA}{dq}\big(q(t_n)\big) \frac{dq_A}{dt}[0; q(t_n)] + \mathcal{O}\big((\Delta t)^3\big),$$

which can be simplified using Eq. (6) to get

$$\int_0^{\Delta t} A\big(q_A[\eta; q(t_n)]\big)\, d\eta = \Delta t A\big(q(t_n)\big) + \frac{(\Delta t)^2}{2}\left(\frac{dA}{dq} A\right)\bigg|_{q=q(t_n)} + \mathcal{O}\big((\Delta t)^3\big). \tag{B4}$$

The expansions of the integrals in $lte^{\text{Rev}}_A$ are now combined to find

$$lte^{\text{Rev}}_A = \frac{(\Delta t)^2}{2}\left(-\frac{dA}{dq}(B + C)\right) + \mathcal{O}\big((\Delta t)^3\big),$$

which is equivalent to the expression in Eq. (29). For $lte^{\text{Ref}}_B$, the second integral is expanded using Eq. (B2). For the first integral in $lte^{\text{Rev}}_B$, use Eq. (A1), with $F = B$, $q = q_B$, and $\phi = q(t_n)$ so that $F\big(q[\eta; \phi]\big) = B\big(q_B[\eta; \phi]\big)$, to find

$$\int_0^{\Delta t} B\big(q_B[\eta; q(t_n)]\big)\, d\eta = \Delta t B\big(q(t_n)\big) + \frac{(\Delta t)^2}{2} \frac{dB}{dq}\big(q(t_n)\big) \frac{dq_B}{dt}[0; q(t_n)] + \mathcal{O}\big((\Delta t)^3\big)$$

which can be simplified using Eq. (6) to get

$$\int_0^{\Delta t} B\big(q_B[\eta;q(t_n)]\big)\,d\eta = \Delta t B\big(q(t_n)\big) + \frac{(\Delta t)^2}{2}\left(\frac{dB}{dq}B\right)\bigg|_{q=q(t_n)} + \mathcal{O}\big((\Delta t)^3\big). \tag{B5}$$

The expansion of the integrals in $lte_B^{\text{Rev}}$ are now combined to find

$$lte_B^{\text{Rev}} = \frac{(\Delta t)^2}{2}\left(-\frac{dB}{dq}(A+C)\right) + \mathcal{O}\big((\Delta t)^3\big),$$

which is equivalent to the expression in Eq. (30). For $lte_C^{\text{Rev}}$, the second integral is expanded using Eq. (B3). For the first integral in $lte_C^{\text{Rev}}$, we use Eq. (A1) with $F=C$, $q=q_C$, and $\phi = q(t_n)+\Delta t(A^*+B^*)$ so that $F\big(q[\eta;\phi]\big) = C\big(q_C\big[\eta;q(t_n)+\Delta t(A^*+B^*)]\big)$, to find

$$\int_0^{\Delta t} C\big(q_C\big[\eta;q(t_n)+\Delta t(A^*+B^*)\big]\big)\,d\eta = \Delta t C\big(q(t_n)+\Delta t(A^*+B^*)\big)$$

$$+ \frac{(\Delta t)^2}{2}\frac{dC}{dq}\big(q(t_n)+\Delta t(A^*+B^*)\big)\frac{dq_C}{dt}[0;q(t_n)+\Delta t(A^*+B^*)] + \mathcal{O}\big((\Delta t)^3\big),$$

which can be simplified using Eq. (6) to get

$$\int_0^{\Delta t} C\big(q_C\big[\eta;q(t_n)+\Delta t(A^*+B^*)\big]\big)\,d\eta = \Delta t C\big(q(t_n)+\Delta t(A^*+B^*)\big) + \frac{(\Delta t)^2}{2}\left(\frac{dC}{dq}C\right)\bigg|_{q=q(t_n)+\Delta t(A^*+B^*)} + \mathcal{O}\big((\Delta t)^3\big)$$

$$\tag{B6}$$

To continue the expansion, use Eq. (B4) and Eq. (B5) to see that

$$q(t_n)+\Delta t(A^*+B^*) = q(t_n) + \int_0^{\Delta t} A\big(q_A[\eta;q(t_n)]\big)\,d\eta + \int_0^{\Delta t} B\big(q_B[\eta;q(t_n)]\big)\,d\eta$$

$$= q(t_n) + \Delta t\Big(A\big(q(t_n)\big)+B\big(q(t_n)\big)\Big) + \mathcal{O}\big((\Delta t)^2\big),$$

which gives

$$\Delta t C\big(q(t_n)+\Delta t(A^*+B^*)\big) = \Delta t C\big(q(t_n)\big) + (\Delta t)^2\left(\frac{dC}{dq}(A+B)\right)\bigg|_{q=q(t_n)} + \mathcal{O}\big((\Delta t)^3\big)$$

and

$$\frac{(\Delta t)^2}{2}\left(\frac{dC}{dq}C\right)\bigg|_{q=q(t_n)+\Delta t(A^*+B^*)} = \frac{(\Delta t)^2}{2}\left(\frac{dC}{dq}C\right)\bigg|_{q=q(t_n)} + \mathcal{O}\big((\Delta t)^3\big).$$

This allows the simplification of Eq. (B6) to

$$\int_0^{\Delta t} C\big(q_C\big[\eta;q(t_n)+\Delta t(A^*+B^*)\big]\big)\,d\eta = \Delta t C\big(q(t_n)\big) + (\Delta t)^2\left(\frac{dC}{dq}(A+B)\right)\bigg|_{q=q(t_n)} + \frac{(\Delta t)^2}{2}\left(\frac{dC}{dq}C\right)\bigg|_{q=q(t_n)} + \mathcal{O}\big((\Delta t)^3\big).$$

The expansion of the integrals in $lte_C^{\text{Rev}}$ are now combined to find

$$lte_C^{\text{Rev}} = \frac{(\Delta t)^2}{2} \left( \frac{dC}{dq}(A+B) \right) \bigg|_{q=q(t_n)} + \mathcal{O}((\Delta t)^3),$$

which is equivalent to the expression in Eq. (31).

## B2 Original splitting

The mapping in the original splitting described in Eq. (20) can be written as

$$\mathcal{F}_{\Delta t}^{\text{Ori}}\big(q(t_n)\big) = q_C \left[ \Delta t; q_B \left[ \Delta t; q_A[\Delta t; q(t_n)] \right] \right] = q_B \left[ \Delta t; q_A[\Delta t; q(t_n)] \right] + \int_0^{\Delta t} C\left( q_C \left[ \eta; q_B \left[ \Delta t; q_A[\Delta t; q(t_n)] \right] \right] \right) d\eta$$

$$= q_A[\Delta t; q(t_n)] + \int_0^{\Delta t} B\big(q_B \left[ \eta; q_A[\Delta t; q(t_n)] \right]\big)\, d\eta + \int_0^{\Delta t} C\left( q_C \left[ \eta; q_B \left[ \Delta t; q_A[\Delta t; q(t_n)] \right] \right] \right) d\eta$$

$$= q(t_n) + \int_0^{\Delta t} A\big(q_A[\eta; q(t_n)]\big)\, d\eta + \int_0^{\Delta t} B\big(q_B \left[ \eta; q_A[\Delta t; q(t_n)] \right]\big)\, d\eta + \int_0^{\Delta t} C\left( q_C \left[ \eta; q_B \left[ \Delta t; q_A[\Delta t; q(t_n)] \right] \right] \right) d\eta.$$

Thus, the local truncation error is expressed as

$$\mathcal{F}_{\Delta t}^{\text{Ori}}\big(q(t_n)\big) - q(t_{n+1})\, d\eta = \underbrace{\int_0^{\Delta t} A\big(q_A[\eta; q(t_n)]\big) - \int_0^{\Delta t} A\big(q[\eta; q(t_n)]\big)\, d\eta}_{lte_A^{\text{Ori}}}$$

$$+ \underbrace{\int_0^{\Delta t} B\big(q_B \left[ \eta; q_A[\Delta t; q(t_n)] \right]\big)\, d\eta - \int_0^{\Delta t} B\big(q[\eta; q(t_n)]\big) d\eta}_{lte_B^{\text{Ori}}} \tag{B7}$$

$$+ \underbrace{\int_0^{\Delta t} C\left( q_C \left[ \eta; q_B \left[ \Delta t; q_A[\Delta t; q(t_n)] \right] \right] \right) d\eta - \int_0^{\Delta t} C\big(q[\eta; q(t_n)]\big) d\eta}_{lte_C^{\text{Ori}}}$$

Note that $lte_A^{\text{Ori}}$ is equivalent to $lte_A^{\text{Rev}}$, which has already been derived in Sect. B1 and is equivalent to the expression in Eq. (25). For $lte_B^{\text{Ori}}$, the second integral is expanded using Eq. (B2). For the first integral in $lte_B^{\text{Ori}}$, use Eq. (A1), with $F = B$, $q = q_B$, and $\phi = q_A[\Delta t; q(t_n)]$ so that $F\big(q[\eta; \phi]\big) = B\big(q_B \left[ \eta; q_A[\Delta t; q(t_n)] \right]$, to find

$$\int_0^{\Delta t} B\big(q_B \left[ \eta; q_A[\Delta t; q(t_n)] \right]\big)\, d\eta = \Delta t B\big(q_A[\Delta t; q(t_n)]\big) + \frac{(\Delta t)^2}{2} \frac{dB}{dq}\big(q_A[\Delta t; q(t_n)]\big) \frac{dq_B}{dt} \left[ 0; q_A[\Delta t; q(t_n)] \right] + \mathcal{O}((\Delta t)^3)$$

which can be simplified using Eq. (6) to get

$$\int_0^{\Delta t} B\big(q_B \left[ \eta; q_A[\Delta t; q(t_n)] \right]\big)\, d\eta = \Delta t B\big(q_A[\Delta t; q(t_n)]\big) + \frac{(\Delta t)^2}{2} \left( \frac{dB}{dq}B \right) \bigg|_{q=q_A[\Delta t; q(t_n)]} + \mathcal{O}((\Delta t)^3).$$

We can now use the expansion of $q_A[\Delta t; q(t_n)]$ in Eq. (A9) to simplify further:

$$\int_0^{\Delta t} B\big(q_B\big[\eta; q_A[\Delta t; q(t_n)]\big]\big)\, d\eta = \Delta t B\big(q(t_n)\big) + (\Delta t)^2 \left(\frac{dB}{dq}A\right) + \frac{(\Delta t)^2}{2}\left(\frac{dB}{dq}B\right)\bigg|_{q=q(t_n)} + \mathcal{O}((\Delta t)^3).$$

The expansions of the integral in $lte_B^{\text{Ori}}$ are now combined to find

$$lte_B^{\text{Ori}} = \frac{(\Delta t)^2}{2}\left(\frac{dB}{dq}(A-C)\right)\bigg|_{q=q(t_n)} + \mathcal{O}((\Delta t)^3),$$

which is equivalent to the expression in Eq. (26). For $lte_C^{\text{Ori}}$, the second integral is expanded using Eq. (B3). For the first integral in $lte_C^{\text{Ori}}$, use Eq. (A1), with $F=C$, $q=q_C$, and $\phi = q_B\big[\Delta t; q_A[\Delta t; q(t_n)]\big]$ so that $F\big(q[\eta;\phi]\big) = C\left(q_C\Big[\eta; q_B\big[\Delta t; q_A[\Delta t; q(t_n)]\big]\Big]\right)$, to find

$$\int_0^{\Delta t} C\left(q_C\Big[\eta; q_B\big[\Delta t; q_A[\Delta t; q(t_n)]\big]\Big]\right) d\eta = \Delta t C\big(q_B\big[\Delta t; q_A[\Delta t; q(t_n)]\big]\big)$$

$$+ \frac{(\Delta t)^2}{2}\frac{dC}{dq}\big(q_B\big[\Delta t; q_A[\Delta t; q(t_n)]\big]\big)\frac{dq_C}{dt}\Big[0; q_B\big[\Delta t; q_A[\Delta t; q(t_n)]\big]\Big]$$

$$+ \mathcal{O}\Big((\Delta t)^3\Big)$$

which can be simplified using Eq. (6) to get

$$\int_0^{\Delta t} C\left(q_C\Big[\eta; q_B\big[\Delta t; q_A[\Delta t; q(t_n)]\big]\Big]\right) d\eta = \Delta t C\big(q_B\big[\Delta t; q_A[\Delta t; q(t_n)]\big]\big)$$

(B8)

$$+ \frac{(\Delta t)^2}{2}\left(\frac{dC}{dq}C\right)\bigg|_{q=q_B\big[\Delta t; q_A[\Delta t; q(t_n)]\big]} + \mathcal{O}\Big((\Delta t)^3\Big)$$

To continue the expansion, use

$$q_B\big[\Delta t; q_A[\Delta t; q(t_n)]\big] = q_B\big[0; q_A[\Delta t; q(t_n)]\big] + \Delta t \frac{dq_B}{dt}\big[0; q_A[\Delta t; q(t_n)]\big] + \mathcal{O}\Big((\Delta t)^2\Big)$$

$$= q_A[\Delta t; q(t_n)] + \Delta t B\big(q_A[\Delta t; q(t_n)]\big) + \mathcal{O}\Big((\Delta t)^2\Big)$$

to get

$$\Delta t C\big(q_B\big[\Delta t; q_A[\Delta t; q(t_n)]\big]\big) = \Delta t C\big(q_A[\Delta t; q(t_n)]\big) + (\Delta t^2)\left(\frac{dC}{dq}B\right)\bigg|_{q=q_A[\Delta t; q(t_n)]} + \mathcal{O}\Big((\Delta t)^3\Big)$$

and

$$\frac{(\Delta t)^2}{2}\left(\frac{dC}{dq}C\right)\bigg|_{q=q_B\big[\Delta t; q_A[\Delta t; q(t_n)]\big]} = \frac{(\Delta t)^2}{2}\left(\frac{dC}{dq}C\right)\bigg|_{q=q_A[\Delta t; q(t_n)]} + \mathcal{O}\Big((\Delta t)^3\Big).$$

We can use the expansion of $q_A[\Delta t; q(t_n)]$ in Eq. (A9) to further simplify the above terms to:

$$\Delta t C\big(q_B\big[\Delta t; q_A[\Delta t; q(t_n)]\big]\big) = \Delta t C\big(q(t_n)\big) + (\Delta t)^2 \left(\frac{dC}{dq} A\right)\bigg|_{q=q(t_n)} + (\Delta t^2)\left(\frac{dC}{dq} B\right)\bigg|_{q=q(t_n)} + \mathcal{O}\Big((\Delta t)^3\Big)$$

$$= \Delta t C\big(q(t_n)\big) + (\Delta t)^2 \left(\frac{dC}{dq}(A+B)\right)\bigg|_{q=q(t_n)} + \mathcal{O}\Big((\Delta t)^3\Big)$$

and

$$\frac{(\Delta t)^2}{2}\left(\frac{dC}{dq} C\right)\bigg|_{q=q_B\big[\Delta t; q_A[\Delta t; q(t_n)]\big]} = \frac{(\Delta t)^2}{2}\left(\frac{dC}{dq} C\right)\bigg|_{q=q(t_n)} + \mathcal{O}\Big((\Delta t)^3\Big).$$

This allows the simplification of Eq. (B8) to

$$\int_0^{\Delta t} C\left(q_C\Big[\eta; q_B\big[\Delta t; q_A[\Delta t; q(t_n)]\big]\Big]\right) d\eta = \Delta t C\big(q(t_n)\big) + \frac{(\Delta t)^2}{2}\left(\frac{dC}{dq} C\right)\bigg|_{q=q(t_n)} + (\Delta t)^2\left(\frac{dC}{dq}(A+B)\right)\bigg|_{q=q(t_n)}$$

$$+ \mathcal{O}\Big((\Delta t)^3\Big).$$

The expansion of the integrals in $lte_C^{\mathrm{Ori}}$ are now combined to find

$$lte_C^{\mathrm{Ori}} = (\Delta t)^2\left(\frac{dC}{dq}(A+B)\right)\bigg|_{q=q(t_n)} + \mathcal{O}\Big((\Delta t)^3\Big),$$

which is equivalent to the expression in Eq. (27).

*Author contributions.* CJV proposed the idea of using the semi-discrete framework for analyzing process coupling problems in EAM, derived the splitting errors presented, and contributed to the interpretation of results from a mathematical perspective. HW proposed and implemented the revised coupling method for aerosol processes in EAMv1, produced the figures of EAMv1 results shown in the paper, and contributed to the interpretation of results from an application perspective. CSW contributed to the application of the framework and interpretation of results from a mathematical perspective. QMB contributed to the derivation of the three-process splitting errors presented. All authors contributed, to varying degrees, to the writing and proofreading.

*Competing interests.* The authors declare that no competing interests are present.

*Acknowledgements.* The authors thank Kai Zhang for conducting parts of the EAMv1 simulations and for providing helpful feedback on the presentation of this paper. The authors thank Sean Patrick Santos for a thorough proofreading of the presentation and for pointing out the link to and distinction from the work by Williamson (2013). Philip J. Rasch is thanked for his continuous encouragement and support for the work. This study was supported by the U.S. Department of Energy's (DOE's) Scientific Discovery Through Advanced Computing (SciDAC) program via a partnership in Earth system model development between DOE's Biological and Environmental Research

(BER) program and Advanced Scientific Computing Research (ASCR) program. The study used resources of the National Energy Research Scientific Computing Center (NERSC), a DOE Office of Science User Facility located at Lawrence Berkeley National Laboratory, operated under Contract No. DE-AC02-05CH11231, using NERSC awards ASCR-ERCAP0025451. Computational resources were also provided by the Compy supercomputer operated for BER by the Pacific Northwest National Laboratory (PNNL). PNNL is operated for DOE by the

Battelle Memorial Institute under contract DE-AC06-76RLO 1830. The work by Lawrence Livermore National Laboratory was performed under the auspices of the U.S. Department of Energy under Contract DE-AC52-07NA27344. LLNL-JRNL-850080-DRAFT.

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
