# Peer review of "Numerical coupling of aerosol emissions, dry removal, and turbulent mixing in the E3SM Atmosphere Model version 1 (EAMv1), part II: a semi-discrete error analysis framework for assessing coupling schemes"

_EGUsphere, 2023_

## Author Response (AR1)

**Point-by-point Responses to Referee Comments**

First and foremost, we thank the referees for their time and effort in reviewing our work, noting ways that the manuscript could be improved, posing some important questions, and catching a number of grammatical mistakes. With their suggestions, we believe the revised manuscript is an improvement upon the original submission. The referee comments and author responses below are more-or-less taken from the online discussion section found at https://egusphere.copernicus.org/preprints/2023/egusphere-2023-1356/.

Following the instructions from Copernicus, this document provides a point-by-point response to all referee comments, noting all corresponding changes in the revised manuscript. The responses follow the sequence: (1) comments from referees, (2) author's response, (3) author's changes in manuscript. Line numbers herein refer to the those in the "Author's track changes file", which has marked the changes from the original submission.

**1 Responses to Referee 1 Comments**

**1.1 Point 1**

**1.1.1 comment**

The idea of using the Taylor series expansion to calculate the LTE for splitting methods on a template equation is well-established. However, compared to similar works available in the literature (properly mentioned in the manuscript), the mathematical apparatus presented here nicely separates the error due to the coupling from the truncation error stemming from the integration of individual processes. I find that the further distinction between isolation-induced and input-induced errors can provide an intuitive understanding also to modelers who might lack a sound mathematical background. Moreover, the framework is particularly flexible, as it can account for an arbitrary number of processes and it does not make any assumption on the nature of the processes (although some degree of regularity is required to apply fundamental results from calculus).

**1.1.2 response**

We are excited that the referee agrees that the framework has the potential to help modelers who lack the numerical analysis background required to readily digest the typical presentation of well-established error analysis.

**1.1.3 changes**

No related changes.

**1.2 Point 2**

**1.2.1 comment**

Nonetheless, I'm a bit skeptical about the direct application of the semi-discrete framework to a full-discrete model. Indeed, the assumption that individual processes are integrated exactly is clearly not met in reality, and the overall error introduced by numerical time-integrators could compensate and/or hide the error associated with the coupling. It follows that even if the framework indicates that the total LTE shrinks when adopting a certain splitting algorithm, the actual model solution might not improve as expected if the splitting error is much smaller than the total error injected by the time-integrators. Therefore, the semi-discrete framework could fail in providing a comprehensive guidance to the choice of the coupling. I would be curious to read the authors' opinion on this point.

**1.2.2 response**

We agree that LTE caused by splitting is only one contributor to the overall error in a fully discretized system. In situations where the time integration of the individual processes is the dominant source of error, one should not necessarily expect reducing a process splitting LTE to result in an observably more accurate solution. Comprehensive guidance to the choice of the coupling should ideally include consideration of all these factors, and we believe a crucial yet missing piece towards obtaining such a comprehensive guide is an analysis on the errors caused by splitting alone. As such, the manuscript is focused on the splitting error. We note that the dust life cycle problem gives a good example of the role splitting error can play. The earlier studies by Wan et al. (2021, doi: 10.5194/gmd-14-1921-2021) and Santos et al. (2021, doi: 10.1029/2020MS002359) showed empirical evidence of process splitting being a major error source in various clouds regimes in EAMv1. In a broader context, the review paper by Gross et al. (2018, MWR, doi: 10.1175/MWR-D-17-0345.1) points out that process splitting/coupling has been a largely overlooked topic in the development of weather, climate, and Earth system models.

   The framework presented in this manuscript can be extended to include time integration errors from individual processes (by replacing exact integrals with discrete sums reflecting quadratures specific to the process integration methods). Such a more comprehensive analysis can be performed when the time integration methods used in individual processes are sufficiently documented, and we are generally interested in performing such analyses for EAM. When performing the more complete analysis, we imagine that the results about splitting error will be useful building blocks, similar to how the distinction between isolation-induced and input-induced errors can provide an intuitive understanding of the LTE caused by splitting.

**1.2.3 changes**

We have included some additional discussion in the revised manuscript:

- some rewording in the abstract on page 1, lines 6-10

- additional discussion in and rearrangement of the introduction on pages 2 and 3, lines 48-52 and 56-62, as well as some minor wording changes on line 64 and 67

- an additional sentence in the conclusion on page 20, lines 435-436.

**1.3 Point 3**

**1.3.1 comment**

Regarding the specific case of the dust life cycle modeling in EAMv1, the total LTE for the original and revised coupling is provided in (28) and (32), respectively. The two expressions only differ for the LTE associated with dry removal (process B). By inspection of the sign of A, C and the derivative of B, it is shown that the magnitude of LTE(B) shrinks under the revised coupling method. However, from a pure mathematical perspective, the total LTE cannot be said to decrease in absolute terms without knowing the sign of LTE(A) and LTE(C). I would encourage the authors to provide some insight on the sign of the terms involved in LTE(A) and LTE(C) to better support their claim.

**1.3.2 response**

The referee's comment highlighted the need to better clarify in the revised manuscript that we are not claiming the revised coupling scheme has a smaller total LTE; rather, we intend to claim the revised coupling scheme has a smaller LTE(B). Specifically, the referee's comment uncovered a collection of typos where we accidentally notated $lte^{\mathrm{Ori}}$ and $lte^{\mathrm{Rev}}$ instead of $lte_B^{\mathrm{Ori}}$ and $lte^{\mathrm{Rev}}$, respectively. We apologize for the confusion and thank the referee for uncovering it. If of interest, more discussion on the merits of reducing LTE(B) in light of other error sources can be found in our online response.

**1.3.3 changes**

We correct a number of typos on page 18, lines 397 - 407, where we accidentally referred to $lte$ instead of $lte_B$. We also added some additional wording on lines 410 and 412 - 414 to emphasize our focus on $lte_B$.

**1.4 Point 4**

**1.4.1 comment**

Moreover, I would also invite the authors to consider including in the framework the actual time-integrators used in the model, so as to come up with an estimate of the total LTE that is better tailored to the use case.

**1.4.2 response**

We agree that including the fully-discrete analysis would very nicely complement the semi-discrete analysis. At this time, a fully-discrete analysis is difficult due to the limited documentation on the integration approaches used for the individual processes. For this reason, we have elected to focus our manuscript on the semi-discrete results. That said, all the suggestions from the reviewer are well taken. In our view, the two companion papers have made an initial attempt to address one small (albeit important) part of a much larger and complex problem. We are continuing to work towards more complete and comprehensive solutions to the coupling problem, and we hope publishing the results from our initial attempts will invite more researchers in the weather, climate, and Earth system modeling communities to give more attention to the coupling challenge.

**1.4.3 changes**

Some commentary added to the introduction as part of the response to (1.2.1).

**1.5 Point 5**

**1.5.1 comment**

In view of future revisions, the authors may want to consider including line numbers in the manuscript.

**1.5.2 response**

The comment about line numbers is well taken: we did not realize that EGU would point to our existing pre-print on arXiv (which forbids line numbers in submissions) instead of hosting a line-numbered version on their own servers, and we will therefore consider a different preprint server that does allow line numbers for future EGU submissions.

**1.5.3 changes**

No related changes.

**1.6 Bulleted Points**

**1.6.1 comment**

1. P5, L4: Delete comma after "time".

2. P5, Eq. (2): Readers without a proper mathematical background may wonder where the expression for the propagated error comes from. It might be worth mentioning that this is a well-known result of calculus.

3. P6, L1: Revise typesetting of $q_{X_i}$. Same comment applies to any following occurrence of this term.

4. P9, Eq. (9): Please introduce the abbreviation LTE in Eq. (2).

5. P14, Fig. 2, box b2: "$y_C^{n+1}$" should be "$q_C^{n+1}$".

6. P15, L15: "the focus here is on the local truncation error"

7. P17, right above Eq. (34): "location" should be "local".

8. P19, L2-3: "The combination of isolation-induced errors leads to an underestimate of the influence of one process on the other."

9. P20, Sect. A1: Could be worth recalling the notation employed in Sect. 2, i.e. that q is a function of time and $\phi$ is the initial condition.

10. P23, right above Eq. (A10): "This allows for the simplification of (A8)".

11. P24, right above Eq. (B1): "solving" repeated twice.

**1.6.2  response**

We thank the referee for catching these grammatical and notation mistakes. All changes have been made with the exception of the wording suggestion in item 6, where the authors prefer the original wording.

**1.6.3  changes**

1. change made on page 5, line 137

2. noted the mean value theorem on page 5, line 147

3. revised typesetting on page 6, line 158, page 7, lines 173-175, page 10, lines 231 and 232, and on page 12, lines 281 and 283

4. change made on page 5, line 146

5. change made on page 14, Figure 2 (b2)

6. no change made

7. change made on page 17, line 386

8. change made to page 19, line 425

9. line added on page 20, liens 456-457

10. change made to page 24, line 517

11. change made to page 24, line 529

**2 Responses to Referee 2 Comments**

**2.1 Point 1**

**2.1.1 comment**

The manuscript focuses on the error related to aerosol emissions in the E3SM Atmosphere Model v1 (EAMv1) but could be extended to other inter-physics coupling with EAM or other similar models. The work presented here provides a valuable tool to climate modelers in a) predicting the numerical error in their models and b) informing decisions about which coupling technique to use. I applaud the authors in doing the work to come up with a mathematical foundation for what is a complicated problem to solve.

**2.1.2 comment**

We are happy to hear the reviewer finds the work a valuable contribution to the climate community.

**2.1.3 changes**

No related changes.

**2.2 Point 2**

**2.2.1 comment**

The authors make the argument that this mathematical framework is applicable to other inter-physics coupling in models like EAM. This is true, and I think an important result of the research. Could the authors briefly comment on the other factors that go into choosing a coupling technique.

**2.3 response**

We find that multiple factors need to be considered when developing an effective approach to evolving any set of coupled processes together in time. These factors include the time scales of each of the processes, whether the processes together form new time scales, whether the processes depend on each other linearly or nonlinearly, whether one process is highly dependent on another, how accurate the resulting solutions need to be, etc. Different coupling approaches may try to more frequently evaluate some processes relative to others and thus address tighter dependencies of one process on another. These strategies will give rise to differing error terms which, when instantiated for a given problem, will show differing values.

**2.4 changes**

We understand the referee to be requesting a response in the discussion of the manuscript instead of changes to the manuscript itself. As such, no changes were made.

**2.5 Point 3**

**2.5.1 comment**

For example, cloud macrophysics and microphysics are treated as separate models in EAM but are conceptually tightly coupled. Do the authors have a sense of whether or not this framework would adequately calculate the error between sequential vs. parallel splitting of these processes? A similar question could be asked for the coupling between fluid dynamics and the sub-grid physics suite.

**2.5.2 response**

The mathematical framework presented here provides general expressions for the local truncation errors caused by process splitting. These expressions are generally valid regardless of which physical processes our symbols A and B correspond to. On the other hand, in specific coupling problems like cloud macropyhsics and microphysics or resolved fluid dynamics and parameterized physics, the characteristics of the processes (e.g., the signs, magnitudes, and time scales of A, B, dA/dq, and dB/dq) can differ substantially and also vary significantly in time and space. Therefore, the specific situations may be much more complicated than what we saw in the dust life cycle problem, and the errors of sequential and parallel splitting can be highly dependent on the associated details. Understanding the implications of the error estimates instantiated for a specific situation will continue to require a close collaboration between numerical analysts and climate scientists. Coincidentally, we have done some investigations into coupling problems related to clouds in EAM and plan to report on these in separate papers.

**2.5.3 changes**

We understand the referee to be requesting a response in the discussion of the manuscript instead of changes to the manuscript itself. As such, no changes were made.

**2.6 Point 4**

**2.6.1 comment**

There is a minor typo in appendix B first sentence "... solving solving ..."

**2.6.2 response**

We thank the referee for catching this typo.

**2.6.3 changes**

The typo is corrected on page 24, line 529.